# Cooperative atomic motion during shear deformation in metallic glass

Yoshinori Shiihara [1] ✉, Takuya Iwashita [2] ✉, Nozomu Adachi[3], Yoshikazu Todaka [3] & Takeshi Egami [4,5,6] ✉

Elucidating mechanical deformation in glassy materials at the atomic level is challenging due to their disordered atomic structure. Using our frozen-atom analysis of the simulation data, we reveal that anelastic deformation in CuZr metallic glasses is fundamentally driven by cooperative atomic motions of tens of atoms elastically linked to one another, forming trigger groups. They initiate localized rearrangements, which can cascade into plastic flow. These cores show no clear structural or elastic precursors in the initial configuration, challenging the idea that deformation occurs in defective regions. Instead, deformation events are highly stochastic and transient, driven by collective atomic motion. This finding not only reshapes our understanding of glassy material deformation mechanisms but also highlights cooperative motion as a key factor in avalanche-like phenomena governing the behavior of disordered systems across multiple scales.

Unraveling the microscopic mechanisms behind plastic and anelastic deformation in glassy materials, such as metallic glasses, remains an enduring mystery over half a century[1–3]. In crystalline materials, lattice defects, such as dislocations, are carriers of plasticity, playing a crucial role in determining mechanical properties. However, the absence of long-range atomic order in glassy materials makes it difficult to define defects. Moreover, in crystals, dislocation density and pinning effects are highly sensitive to microstructural variations, leading to significant differences in mechanical strength[4,5]. In contrast, metallic glasses display remarkably universal mechanical responses across different compositions and processing conditions[6], suggesting that the conventional paradigm linking structural defects to mechanical behavior may not directly apply to metallic glasses. This raises fundamental questions: if defects are not the primary deformation carriers, what governs mechanical behavior in glassy materials? Identifying these underlying mechanisms could lead to a more comprehensive understanding of deformation behavior in strongly disordered solids.

We show, through computational simulations of shear deformation in metallic glasses, that the fundamental deformation elements in these materials are atomic groups exhibiting cooperative motion. Numerous studies have investigated atomic cooperativity in glasses and supercooled liquids, both under applied stress[7–11] and in stress-free conditions[12–17]. Yet, even in simulations, cooperative movements have thus far been inferred only from the displacements of atomic groups during stress relaxation events. A widely used framework for describing the plastic deformation of metallic glasses is based on shear transformation zones (STZs), which are localized, collective atomic rearrangements[6,7,9,18]. In the framework of self-organized criticality, the interaction and propagation of STZs in an avalanche-like manner have been widely discussed[19–21]. These discussions suggest that plastic deformation is governed by strongly interacting elements exhibiting collective behavior across multiple length scales. However, a detailed microscopic understanding of these processes—particularly the triggers for deformation events and the nature of their interactions—remains incomplete. We identified this cooperative behavior as the essential core of STZs, which we refer to as the STZ core. We analyze their size, structural, and mechanical characteristics, while also revealing the presence of cascading STZ cores that initiate the early stages of avalanches. This analysis enhances the understanding of

[1]Graduate School of Engineering, Toyota Technological Institute, Nagoya, Aichi, Japan. [2]Department of Science and Technology, Oita University, Oita, Japan. [3]Department of Mechanical Engineering, Toyohashi University of Technology, Toyohashi, Aichi, Japan. [4]University of Tennessee, Knoxville, TN, USA. [5]Department of Physics and Astronomy, University of Tennessee, Knoxville, TN, USA. [6]Materials Science and Technology Division, Oak Ridge National Laboratory, Oak Ridge, TN, USA. ✉e-mail: shiihara@toyota-ti.ac.jp; tiwashita@oita-u.ac.jp; egami@utk.edu

deformation mechanisms in metallic glasses and other amorphous materials.

Cooperativity in glassy materials is concealed within their complex atomic dynamics, making its extraction challenging. To address this, we introduce frozen-atom analysis, a computational technique that explores what-if scenarios. In real life, it is impossible to go back in time, but in a simulation, we can trace time back, like in a time-machine, and change history. We identify a local plastic deformation event, and go back in time to the instance just before the start of the event. Then we artificially immobilize an individual atom. If this atom is involved in cooperative motion, the deformation event will not happen when the simulation is continued with the atom immobilized. We applied this conditional freezing to each atom in the system and identified the atoms involved in cooperative motion, thereby revealing the STZ core. Through the comparison of parallel worlds enabled by frozen-atom analysis, we uncover the fundamental nature of STZs, distinct from the conventional perspectives that rely upon the post-mortem analysis of the event. We found that the STZ core consists of tens of atoms, which engage in cooperative motion. This value may differ among various glass systems; nevertheless, the frozen-atom analysis provides an unambiguous determination of it. These atoms lack distinct structural features, suggesting that the STZ core can emerge anywhere. Furthermore, it induces plastic deformation among the surrounding atoms and acts as a trigger, initiating cascades that drive avalanches. These findings represent a drastic departure from the conventional view that STZs correspond to defects or soft spots, paving the way toward deeper insights into the physical principles governing the deformation mechanisms of glassy materials.

## Results

### Frozen-atom analysis

The frozen-atom analysis enables the unambiguous identification of atomic groups involved in cooperative motion, specifically STZ cores. The athermal quasi-static (AQS) method[22,23] is a widely used computational approach for modeling critical slip phenomena under quasi-static shear in glassy materials. In this method, a model glass undergoes incremental affine shear deformation, followed by athermal relaxation, transitioning through mechanically stable configurations. A characteristic feature of this process is the sudden shear stress drop during relaxation, signaling a plastic deformation event involving atomic rearrangements (Fig. 1a). Frozen-atom analysis is applied at this critical stress drop to pinpoint the STZ core. The method starts from the atomic configuration just before the deformation event, applies an affine deformation, and artificially freezes one atom while allowing the rest to relax. If the plastic deformation event does not occur, the frozen atom is identified as a part of the STZ core, confirming its essential role in the deformation event. Figure 1b schematically illustrates that freezing a single atom disrupts the cooperative motion of the entire group, emphasizing their interdependence, although this representation is oversimplified because a three-dimensional glass has much higher degrees of freedom.

We implement this method as follows: during the relaxation process of a stress-drop event in the AQS procedure, one atom from a system of $n$ atoms is selected, and its motion is artificially frozen. The remaining atoms are allowed to relax, and the total displacement of the entire system is recorded. This is referred to as the $D_f$ parameter (in Å) here.

$$D_f = \sqrt{\sum_{i=1}^{N} \| \mathbf{d}_i \|^2}, \qquad (1)$$

where $\mathbf{d}_i$ represents the displacement vector of atom $i$ during the structural relaxation and $N$ denotes the total number of atoms in the simulation box. This process is repeated for every atom, generating a $D_f$ parameter value for each atom. This $D_f$ parameter clearly distinguishes between atoms that participate in cooperative motion and those that do not: a $D_f$ value of zero indicates that freezing the atom's motion disrupts the cooperative displacement, showing that the atom with $D_f = 0$ is a part of the deformation element. Conversely, a non-zero $D_f$ value indicates that the atom's displacement is not critical to the cooperative motion. This method appears similar to the artificial manipulations, such as the atomic pinning approach to probe atomic cooperativity in flow[24,25] or other related interventions[26,27], but our approach is quite different in that atomic freezing is used merely to probe the role of each atom in deformation, and it does not affect the flow itself (see Supplementary Note 1 for details).

In conventional simulations and experiments, pinpointing the atomic-level interactions or mechanisms that drive observed plastic deformation is quite challenging. Here, by deliberately controlling the motion of specific atoms, frozen-atom analysis enables us to virtually modify their movements and directly probe causal relationships between an atom and the flow. Comparing multiple "parallel worlds," where different atoms are immobilized, makes it possible to extract the hidden cooperative motion embedded within the disordered structure of glassy materials.

### Identification of atomic group in cooperative motion

The histogram of the $D_f$ parameter obtained from a stress-drop event is shown in Fig. 1c as an example, from which three groups of atoms can be identified, as illustrated on the right side of the figure. The first group consists of atoms with $D_f$ values close to zero (visualized as $D_f \leq 0.5$ in the figure on the right). As mentioned earlier, these are the atoms frozen during the relaxation process that halt the motion of other atoms. In other words, this group represents atoms critically needed for cooperative motion. As shown in the Methods section, such cooperative motion was observed in all 60 stress relaxation events investigated in this study. Therefore, it can be concluded that all STZ cores observed here are the atomic groups critical for cooperative motion.

The second group of atoms is located near the peak of the histogram. Even when their motion is frozen, the cooperative motion occurs during the relaxation process, meaning they are not directly involved in the activation of the deformation event. The peak position of the histogram corresponds to the $D_f$ value when no atoms are frozen. As shown in the figure on the right ($0.5 < D_f \leq 1.3$ case), most atoms in the system belong to this group. The third group, found in the tail of the histogram and located near the STZ group ($D_f > 1.3$ case), experiences large displacements due to cooperative motion, but this does not affect the deformation event. When their motion is frozen, all other atoms move to reproduce the original displacement, leading to large $D_f$ values for this group, indicating that they do not stop the cooperative motion.

A remarkable feature of the $D_f$ parameter lies in the gap observed in the histogram. As shown in Fig. 1c, this gap clearly distinguishes the group of atoms belonging to the STZ core from other atoms. For comparison, Fig. 1d and e show histograms of conventional indicators: the non-affine displacement measure $D^2_{min}$ and the change in atomic von Mises stress during relaxation. The atoms in the tail of the distribution in Fig. 1d, namely, those with large non-affine deformation, are often assumed to represent the central region of the STZ. Similarly, in Fig. 1e the atoms exhibiting the most pronounced changes in local von Mises stress are also regarded as part of this region. In both cases, the highlighted atoms overlap considerably with those identified by the $D_f$ parameter in Fig. 1c. This indicates that, although conventional descriptors capture atoms involved in the event, the size of the atomic group they identify can vary significantly depending on the chosen threshold, which makes their interpretation ambiguous. In contrast, the frozen-atom analysis provides an unambiguous distinction between the indispensable STZ core (the group close to $D_f$) and the

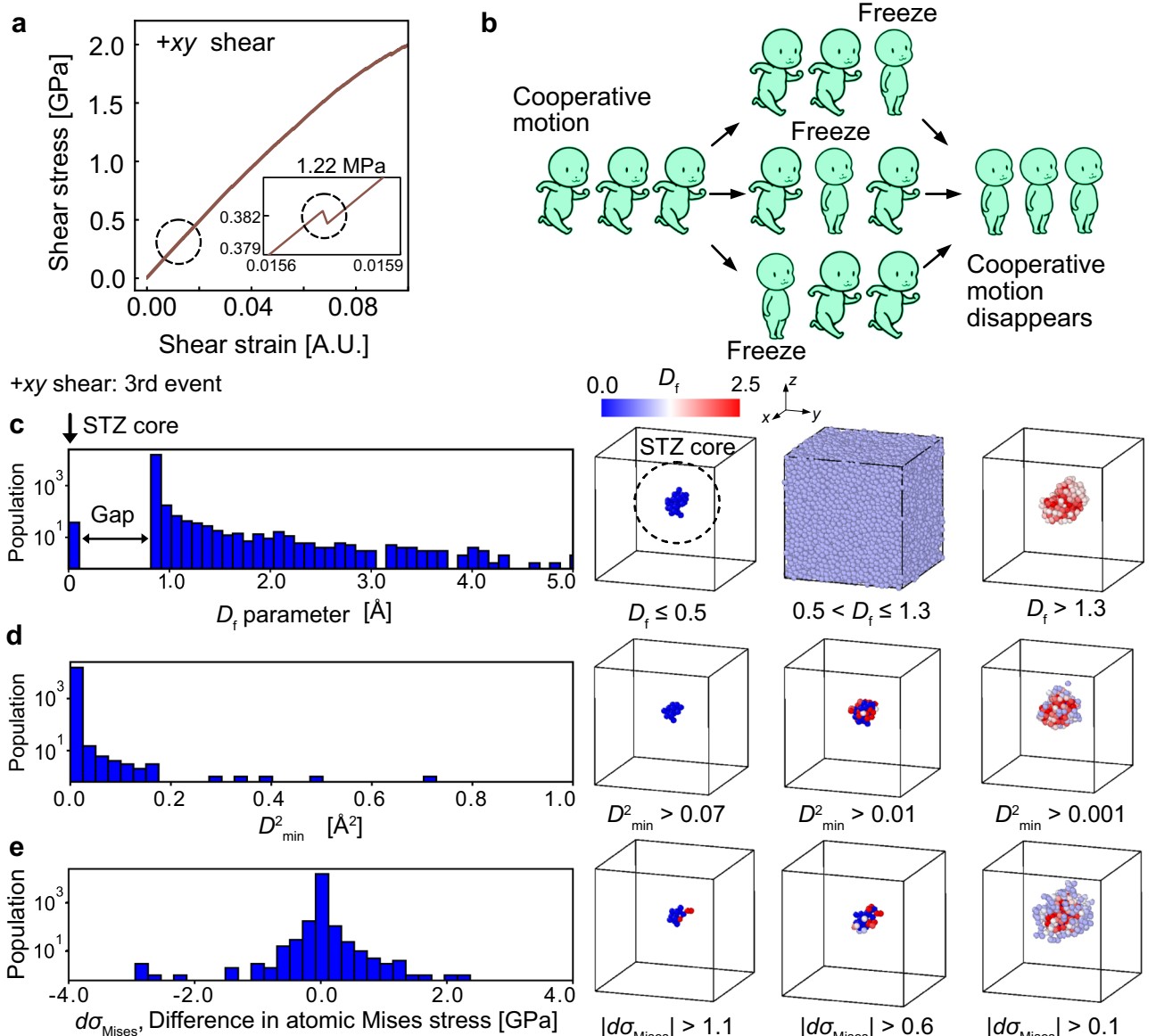

**Fig. 1 | Frozen atom analysis in metallic glass. a** Stress–strain curve of metallic glass obtained from athermal quasi-static (AQS) simulations in −$xy$ shear direction. The plastic event to which frozen-atom analysis is applied is marked with a dashed circle. The inset zooms in on the stress drop associated with the event, showing a decrease of -1.2 MPa. **b** Conceptual illustration of frozen-atom analysis. Cooperative motion is illustrated by a group of people walking together, where freezing one individual restricts others. Stopping the front person blocks those behind, freezing the middle constrains both sides, and immobilizing the back pulls the front individuals backward. This analogy captures the cooperative nature of atomic rearrangements in metallic glasses. **c–e** Logarithmic histograms comparing different shear transformation zone (STZ) descriptors for the first stress relaxation event. **c** The $D_f$ parameter obtained from frozen-atom analysis. **d** The non-affine squared displacement, $D^2_{min}$ **c**. **e** The difference in atomic von Mises stress before and after the event. The right panels visualize atoms with descriptor values in specific ranges using OVITO software[60], where atoms are color-coded based on the $D_f$ parameter.

surrounding atoms. The sizes of the STZ cores obtained in this study were up to 116 atoms, with an average of about 40 atoms, which is consistent with the sizes of STZs reported in previous studies[1]. However, in previous studies, the number of atoms constituting the STZ depends on arbitrary parameters, such as the threshold value. In contrast, the frozen-atom method with the $D_f$ parameter identifies the STZ core without ambiguity.

### Characteristics of STZ core

The characteristics of STZs, including their size, shape, distribution, and hardness, have been subjects of long-standing debate[1,28]. We studied the atomic-level characteristics of the STZ core atoms in the zero-strain initial configuration before any shear was applied, in order to

test whether precursors of STZ cores exist. The nature of the STZs identified in this study differs significantly from those previously discussed. Figure 2a illustrates the spatial distribution of STZ cores detected in the 60 events considered. These STZ cores are widely spread across the system, indicating that STZ cores do not form in a spatially confined region but rather emerge throughout the material. The color scale shows how often each atom was classified as part of an STZ core. The low occurrence of overlaps suggests that these events rarely occur repeatedly in the same location.

Figure 2b plots the features of Voronoi polyhedra, specifically the number of faces and the volume in the initial configuration, for the 100,000 atoms randomly sampled from 960,000 atoms considered in this calculation. The circle size represents the $D_f$ parameter, where

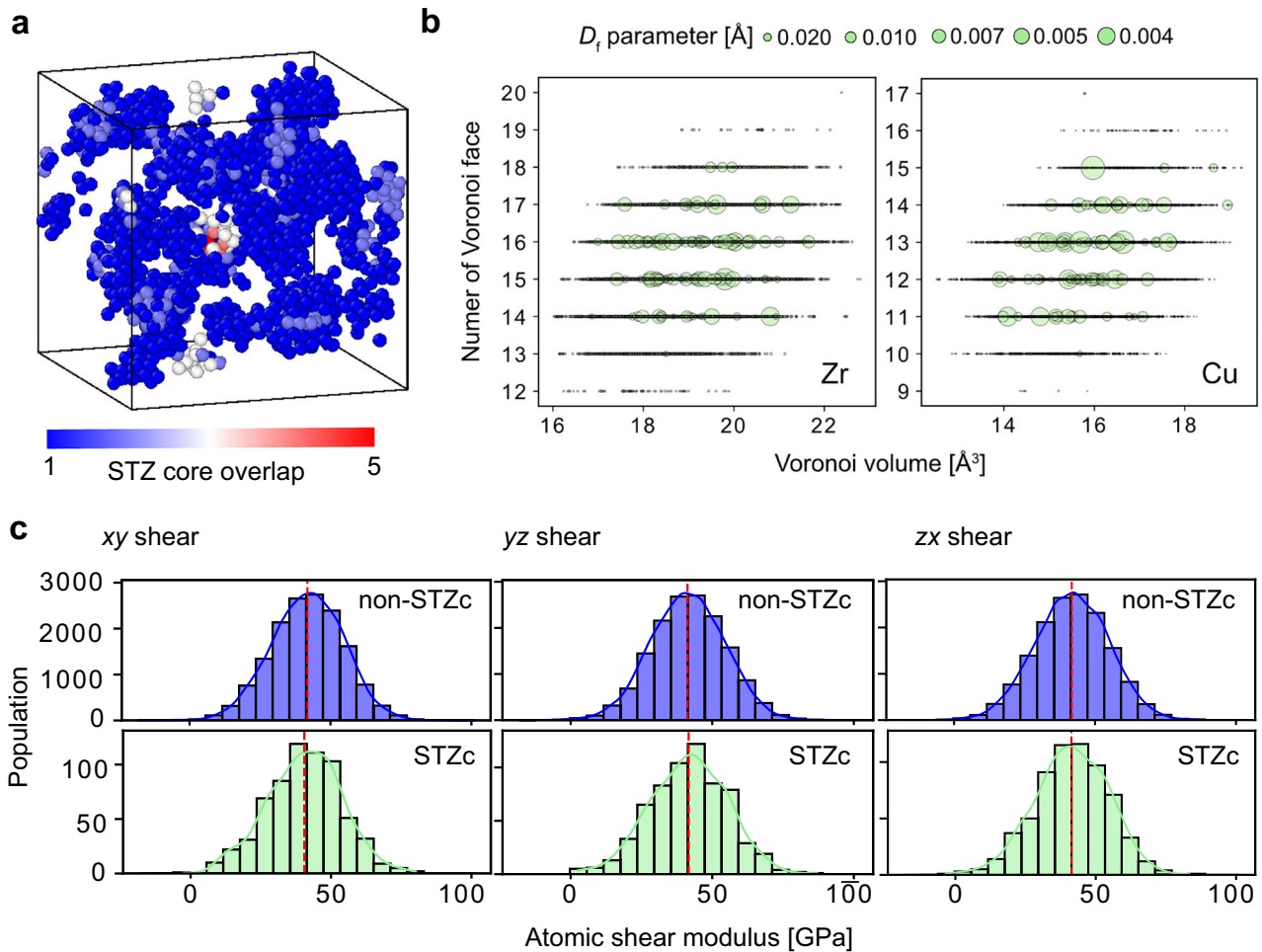

**Fig. 2 | Characteristics of shear transformation zone cores. a** Atomic configurations identified as shear transformation zone (STZ) cores in 60 deformation events. The color represents the frequency with which atoms were classified as STZ cores across multiple events. **b** Comparison of Voronoi features (number of Voronoi faces and Voronoi volume per atom) and $D_f$ parameters for Zr (left) and Cu (right). Larger circles represent atoms with larger $D_f$ values, indicating that they are more involved in cooperative motion. **c** Histograms of the atomic shear modulus for STZ core atoms (STZc) and non-core atoms (non-STZc). The atomic shear modulus corresponding to each shear direction is shown. The STZ cores considered are those driven by deformation along the respective shear direction. The modulus is calculated as the analytic derivative of the embedded atom method (EAM) potential with respect to cell strain[30]. Dashed lines indicate the average values for each distribution.

larger circles basically correspond to atoms identified as participating in cooperative motion. Despite the widespread use of Voronoi polyhedra as descriptors for characterizing the atomic structure of defects, including STZs, the nearly uniform distribution of atoms with low $D_f$ values within the STZs suggests that extracting STZs based solely on Voronoi features may not be a reliable approach. Figure 2c compares histograms of atomic shear modulus at zero strain for STZ core atoms detected in each shear direction with those of non-core atoms. The obtained histograms and modulus values are consistent with those reported in previous studies[29,30]. No significant difference in hardness was observed between STZ core atoms and other atoms. To further examine the elastic properties, we conducted coarse-grained evaluations of atomic elasticity based on the STZ cores. Even in the undeformed configuration, variability was evident: some STZ cores were stiffer than the cell average, while others were softer. The average shear moduli of the STZ cores were $40.5 \pm 3.8$ GPa ($xy$), $40.7 \pm 3.1$ GPa ($yz$), and $42.0 \pm 3.9$ GPa ($zx$), compared with the overall averages of 41.8, 41.9, and 41.7 GPa, respectively. This indicates that STZ cores in the initial state are neither soft spots as traditionally perceived nor hard spots; in other words, they do not exhibit any distinct characteristics in terms of elastic constant.

In the "Methods" section, we present the evolution of the average atomic energy and the von Mises stress within each STZ core during the AQS process. Even within the apparent elastic regime, energy and stress responses exhibit strong variation with applied stress. This indicates that the mechanical response of this system is highly nonlinear, and atoms are possibly influenced by interactions with other atoms within the STZ core and in other activated STZ cores. This underscores the role of intrinsic structural inhomogeneity in driving localized structural and mechanical changes under external stimuli, challenging predictions based on the initial state.

Figure 3 explores the relationship between the STZ core size and the magnitude of stress drops observed in the 60 events considered. The histogram of stress drop magnitude follows a power-law distribution (Fig. 3a), whereas the histogram of the STZ core differs from it, exhibiting a shape similar to a Poisson distribution (Fig. 3b). Whereas it is widely believed that the size of the STZ is proportional to the magnitude of the stress drop, the histograms of stress drop magnitude and STZ core size in this study show otherwise. Even though some weak correlations may be found in Fig. 3c between the magnitude of stress drop and the macroscopic strain at which the event occurred, the size of the STZ core has no correlation with either.

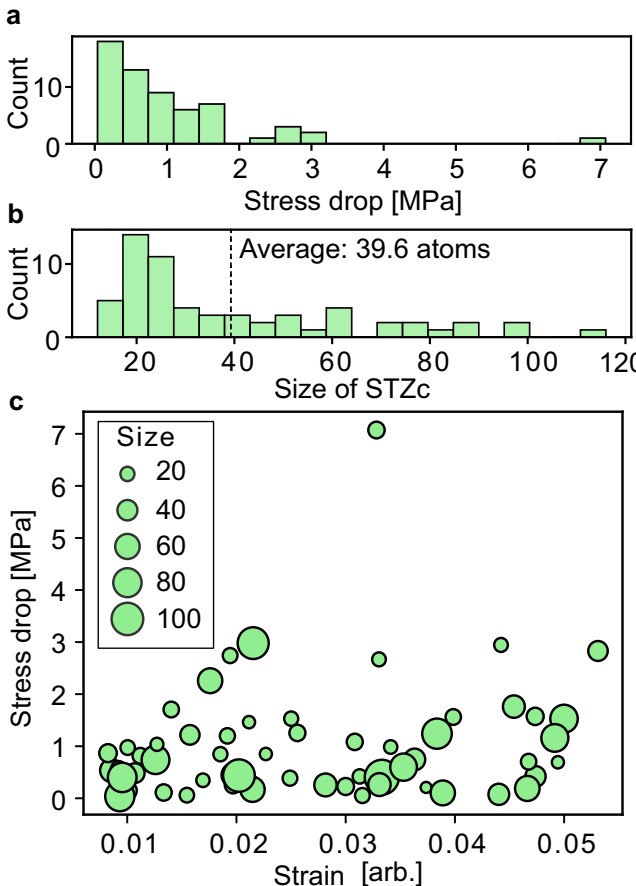

**Fig. 3 | Event size and corresponding shear transformation zone core size.**
**a** Histogram of stress-drop magnitudes for the 60 events considered. **b** Histogram of the number of atoms included in the shear transformation zone core (STZc).
**c** Relationship between the applied shear strain on the cell and the stress-drop magnitude. The size of the circles is proportional to the number of atoms in the STZc observed during each event.

These results suggest that the atoms in STZ cores are not in special environments that could be identified as defects, in agreement with the earlier study[31]. It is likely that any atom can be involved in STZ cores. This may explain why the critical strength is so universal among various metallic glasses[6].

### Local configurational excitation in STZ core

To further examine the nature of the STZ core thus defined at the atomic level, we investigated the local configurational excitation (LCE) that occurs during the stress-drop process. LCE refers to the action of losing or gaining nearest-neighbor atoms around a particular atom. This change in the atomic connectivity network due to bond switching is known to govern the macroscopic viscosity of high-temperature liquids (details are provided in ref. [32] and the "Methods" section). Figure 4a compares the number of bond formations and breakages that occur in a single atom during the stress relaxation process between STZ core atoms and non-STZ core atoms. It was found that most LCEs involving three or more bond breakages occur in atoms belonging to the STZ core.

We observe that many atoms in the close vicinity of the STZ core are also displaced due to the activation of the core, as exemplified by the third atom group with $D_f > 1.3$ in Fig. 1c. LCEs that occur in the STZ core induce non-affine displacements in its surroundings, and these displacements can sometimes drive other STZ cores. Figure 4a shows that some atoms, despite being classified as non-STZ core, exhibit a

significant number of LCEs. Figure 4b illustrates the corresponding event. While the $D_f$ parameter identifies an STZ core, other parameters suggest that other atoms in the vicinity are also involved in the deformation event. Of the atoms involved in LCE, only 60% are included in the STZ core, while the remainder are either surrounding the STZ core or part of events triggered by the core. The LCE analysis revealed that eight out of the 60 events considered exhibited cascade-like behavior (see the "Methods" section for details). These results imply that the activation of one STZ core triggers a local avalanche of deformation, which is more extended and involves a larger number of atoms. The deformation event, or the STZ, is complex and involves the trigger and subsequent large local deformation in its vicinity. This distinction between triggering and triggered STZ regions becomes clear through the frozen-atom analysis, which enables their separation.

## Discussion

The differences in the characteristics between the STZs defined in previous studies and those in this study originate from an important distinction in the STZ concept. In prior research, physical quantities such as $D^2_{min}$, which have been used to identify STZs, primarily reflect the a posteriori consequences of STZ activation. Therefore, STZs are defined as the result of the deformation event rather than the dynamic phenomenon as a whole. Based on this interpretation, it is natural to expect a proportional relationship between the size of the stress drop and the size of the STZ. In contrast, in this study, the STZ core is defined in the a priori state in terms of the eventual contribution to the deformation event. To be specific, STZ cores are defined as atomic groups that relax stress through cooperative motion, and this definition is made for the state prior to the deformation event. The analysis of the characteristics of individual atoms in STZ cores reveals that the formation of STZ cores is a highly stochastic and transient process: unlike conventional deformation units such as dislocations, they do not possess distinct structural features and can form anywhere. It is possible that local factors other than defective atomic structures, such as elastic and strain heterogeneities within a disordered structure, lead to local deformation events. However, the analysis of the STZ core atoms in the undeformed initial state did not reveal any distinctive elastic characteristics. STZs are statistically created and disappear after deformation events, as presumed in the STZ theory by Langer[33].

Furthermore, contrary to the widely spread idea of the STZ as the site of massive liquid-like flow, the number of atomic bond breakings is rather small. Only 32% of atoms in the STZ core are involved in LCE. Instead, the STZ core functions as a trigger, where atoms move cooperatively as an elastically bound group, initiating large-scale deformation. On average, ~40 atoms play a crucial role in deformation, collectively constraining each other's motion. This elastic nature of STZ cores contrasts with the conventional soft spot interpretation, which assumes localized regions of structural weakness. While STZ cores are not hard spots either, their deformation behavior emerges nonlinearly rather than being determined solely by pre-existing spatial heterogeneity. The presence of elastically bound groups in deformation is consistent with our earlier observation of transient solid-like groups of atoms during shear flow[34], further supporting the idea that STZs are distinct from pre-defined defects.

A key finding of this study is that STZ cores do not exhibit unique structural features in their initial state, making them inherently unpredictable based on conventional descriptors such as Voronoi volume, atomic stress state, or atomic shear modulus. Unlike defects that can be identified from static configurations before deformation, STZ cores emerge dynamically through cooperative atomic motion in response to the nonlinear stress and strain behavior of a glassy structure under external stimuli. This suggests that LCEs and the activation of the STZ core are an emergent phenomenon, not predetermined, but occur as a consequence of evolving interactions. Furthermore, one STZ core can drive another, leading to cascading

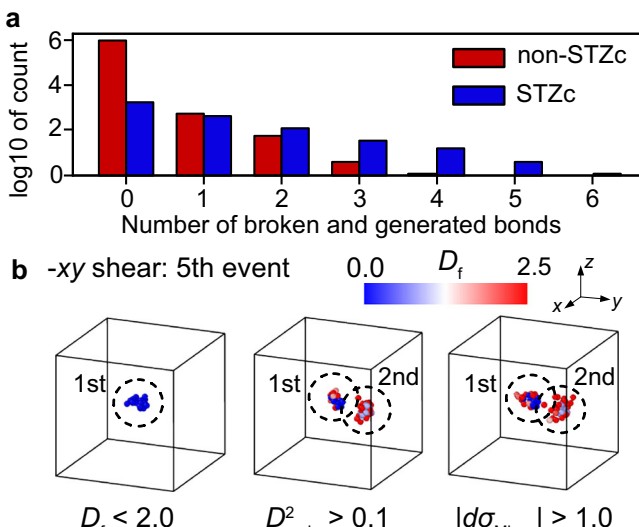

**Fig. 4 | Relationship between local core excitation and shear transformation zone cores. a** A logarithmic histogram for the sum of broken and newly formed bonds for a total of 960,000 atoms involved in the 60 stress relaxation events considered in this study. **b** Visualization of the shear transformation zone core (STZc) that triggered events containing non-STZc atoms with six broken and generated bonds (5th event in *-xy* shear direction with 2.67 MPa stress drop), using each descriptor. Atoms are color-coded based on the $D_f$ parameter.

deformation events. This indicates that deformation is not solely dictated by isolated activation sites but also by interactions between STZ cores and their surrounding environment.

STZ cores, or their specific characteristics, may nevertheless be extractable through topological analyses that incorporate higher-dimensional structural information. Here, this refers to describing atoms collectively rather than individually, consistent with previous reports that collective structural diversity rather than specific local motifs governs the dynamics of amorphous alloys[35,36]. Graph neural networks, which have recently been applied in the context of STZs[37], may capture subtle features not readily identifiable by conventional analysis. Such approaches, however, become meaningful only after the nature of the STZ is identified, and the boundary of the STZ core is clearly defined; frozen-atom analysis provides this definition unambiguously by directly identifying the indispensable atoms whose immobilization blocks the cooperative process.

The nature of the STZ core and its spatial extension were identified by the frozen-atom analysis, which functions like a time machine to explore parallel histories of atomic motion. While conventional methods focus on analyzing characteristic deformation units and lattice defects, this technique enables the systematic isolation of fundamental triggers of plastic deformation by freezing specific atoms and tracing their impact. The scenario presented by this analysis is as follows. First, the STZ core is activated as a unit of cooperative motion. The location of STZ cores is unpredictable. Each STZ core involves atomic bond rearrangement, specifically LCE, which generates non-affine deformation in the surrounding region. This non-affine deformation, in turn, drives other STZ cores, leading to a cascade of deformation events. These characteristics align with the self-organized criticality (SOC) model, where small perturbations occurring anywhere in space naturally evolve into a critical state that triggers cascading events. This process gives rise to avalanche dynamics observed across different scales in various phenomena[20,38–40]. In the mathematical model, the emergence of SOC-like behavior depends on the conditions for trigger formation and their interactions[19]. Identifying STZ cores and their cooperative motion can advance the quantitative understanding of SOC in glassy materials, providing key parameters—such as trigger

size, spatial distribution, external driving conditions, and their interaction—that serve as a foundation for integrating SOC models into material design. Furthermore, considering the universality of avalanche dynamics, our findings suggest that cooperative motion may serve as a fundamental mechanism underlying SOC behavior in a broad range of nonequilibrium systems.

In conclusion, this study reveals that the fundamental deformation unit in metallic glasses is made of two components: a trigger and subsequent deformation. We define this trigger atomic group as the STZ core, which has an average size of approximately 40 atoms. This number may vary across different glass systems and could depend on chemical composition and preparation conditions. However, the frozen-atom analysis can determine this number without ambiguity. This cooperative atomic motion functions as the trigger for local deformation events, involving a larger number of surrounding atoms in the subsequent deformation process. The absence of distinctive local structural characteristics for the STZ core atoms suggests that STZ cores are not structural defects and can form anywhere in response to external stimuli. These findings challenge the conventional understanding of this complex phenomenon and advance the elucidation of deformation mechanisms in glassy materials. From a broader perspective, the concept established in this study—that cooperative motion serves as the trigger for avalanche-like deformation—has the potential to bridge the understanding of plastic deformation in glassy materials with a wide range of nonequilibrium and nonlinear response systems, where avalanche dynamics play a crucial role. The insights gained in this study are expected to serve as a foundation for their further exploration and deeper understanding.

## Methods
### Simulation setup
We conducted quasi-static shear deformation on a periodic three-dimensional metallic glass $Cu_{50}Zr_{50}$, comprising 16,000 atoms. The molecular dynamics simulations were carried out using the LAMMPS software package[41] with the embedded atom method (EAM) potential for Cu-Zr[42]. The metallic glass was prepared by melting and quenching a B2 CuZr alloy. A velocity distribution corresponding to 2000 K, which is well above the glass transition temperature of this alloy (~750 K, see Supplementary Fig. 2), was applied to the B2 structure, and the system was directly cooled to 0 K at a rate of $10^9$ K/s in an NVT ensemble using the Nosé–Hoover thermostat[43,44] without an intermediate annealing period. After cooling, the atomic structure and simulation cell were relaxed using the conjugate gradient method until all stress components converged to below 0.002 MPa. The final simulation box had a characteristic length of ~65.4 Å and was nearly cubic but with slight tilts in the cell vectors, as shown in Supplementary Table 1.

The athermal quasi-static (AQS) method[22,23] was then employed to apply quasi-static shear deformation. This involved applying a small shear strain to the cell vectors of the glass and affinely deforming the atomic structure in accordance with the strain, followed by energy minimization at 0 K to ensure zero force on all atoms. This procedure was repeated 15,000 times to induce deformation until the engineering shear strain reached 15%, with a strain increment of $10^{-5}$ per step. A total of 60 plastic events were sampled, including ten events for each of the six shear directions (±*xy*, ±*yz*, ±*zx*), as shown in Fig. 5.

### Frozen-atom analysis to identify STZ core
Various models, such as the shear transformation zone (STZ) model[6,7,9,18], have been proposed to describe deformation behavior, and a number of attempts have been made to identify the deformation mechanism through molecular dynamics simulations[23,35,37,45–48]. In the typical approach, the AQS method, incremental uniform (affine) shear deformation is applied to a model glass structure, and the atomic structure is relaxed athermally at each deformation step. When the shear stress drops during the relaxation, it is assumed that a plastic

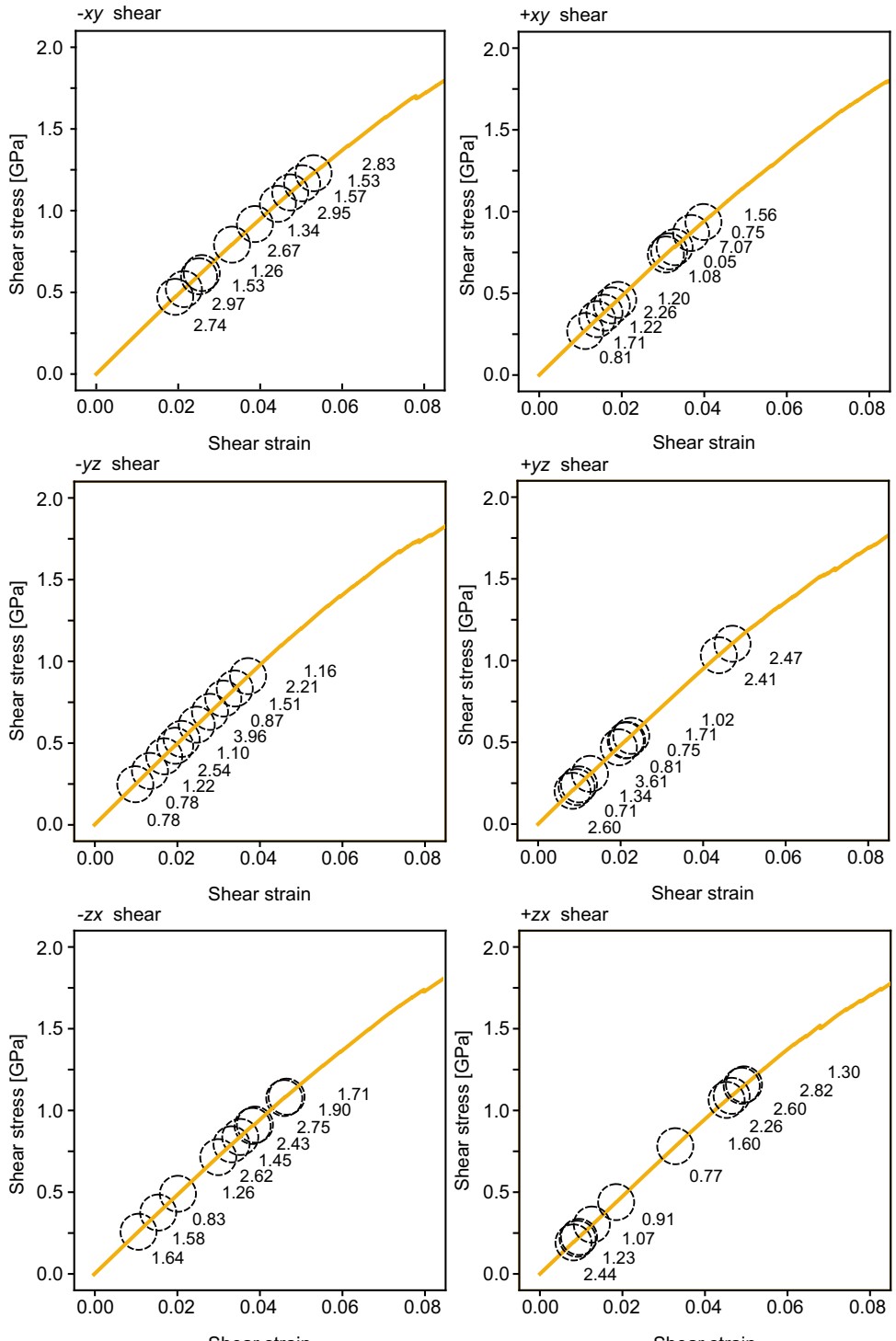

**Fig. 5 | Stress–strain curves obtained by athermal quasi-static simulation.** Each circle represents one of the 60 stress-drop events considered in this study, and the magnitude of the stress drop [MPa] is also described.

deformation event occurs through some rearrangement in atomic configuration. For convenience, we call the area with strong plastic deformation an STZ. The changes in certain geometric or physical quantities before and after the relaxation are used to extract features of local atomic behavior. To identify the STZ, various metrics have been introduced. These include the non-affine squared displacement $(D_{min}^2)$[18,47,49], Voronoi-based metrics[50,51], local stress drops[52], local shear elasticity[53], von Mises strain, atomic displacements[54], and soft modes obtained through Hessian diagonalization[55]. Among these, $D_{min}^2$ has

been particularly widely used. However, this method suffers from significant problems. In defining the non-affine displacement, it is assumed that the affine strain is uniform, whereas the actual stress field is rather inhomogeneous in space. For instance, when an STZ is activated, local shear softening occurs, producing an elastic field around the STZ[2,56]. Furthermore, to identify the STZ, a threshold value for $D_{min}^2$ is set[57]. However, the choice of this value significantly affects the spatial distribution of the deformation elements, as shown later, introducing large ambiguity. In order to elucidate the deformation mechanism, in

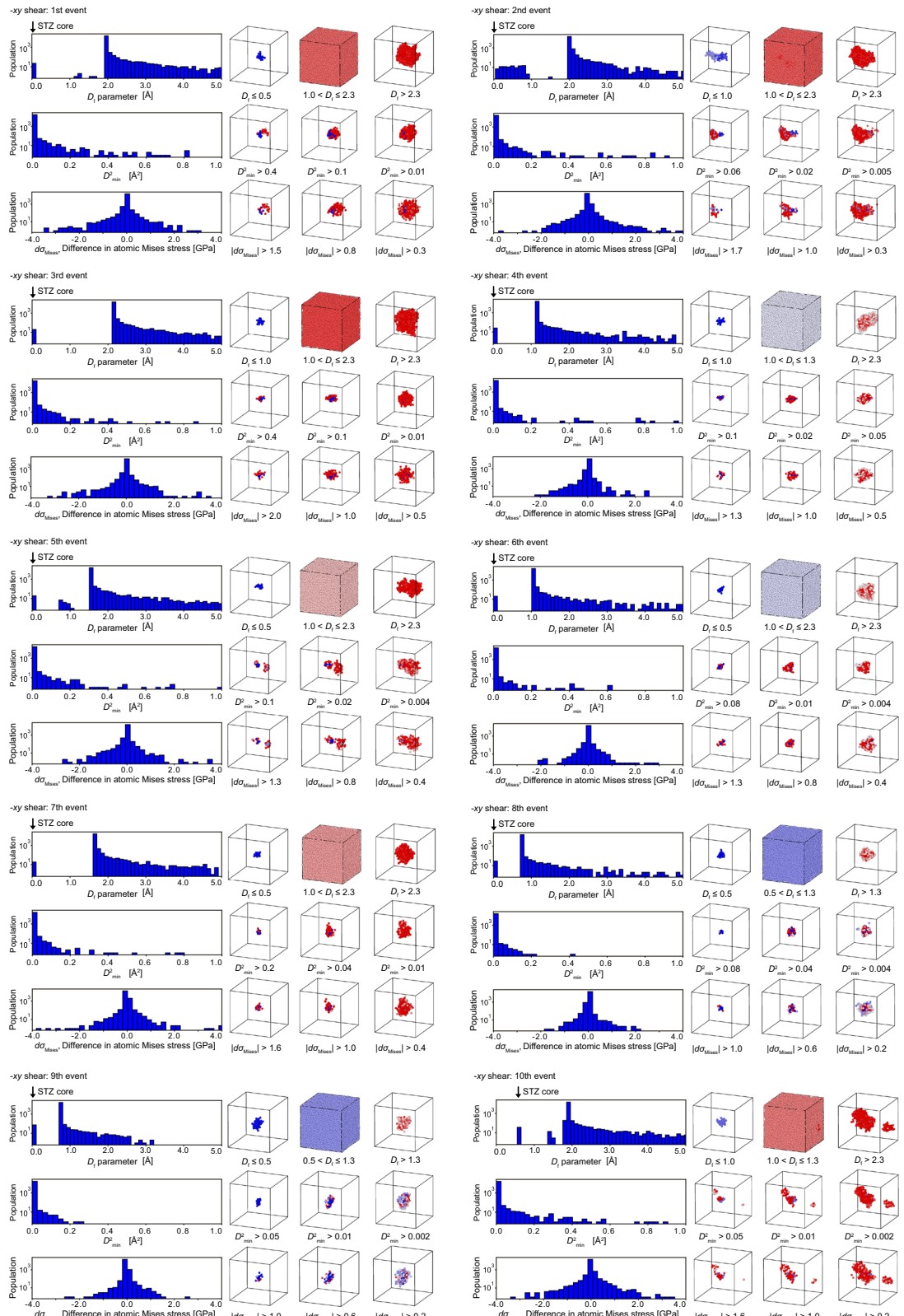

**Fig. 6 | Frozen atom analysis results for ten events in the −xy shear direction.** These results follow the same format as Fig. 1 in the main text. Details of the descriptors and visualization methods are described in the caption of Fig. 1.

this work, we take a different approach and re-examine the definition of the deformation zones, such as STZs.

In the frozen-atom analysis, we start with the atomic configuration just a step before a cooperative deformation event (see Fig. 1a). Then, after applying an affine deformation that will result in the deformation

event, we artificially freeze an atom, and relax the rest during the relaxation process. If the cooperative motion ceases as a result of this freezing, we identify this atom as belonging to the atomic group we define as the STZ core, because the movement of this atom is essential for the deformation event to take place. Figures 6 and 7 present 20

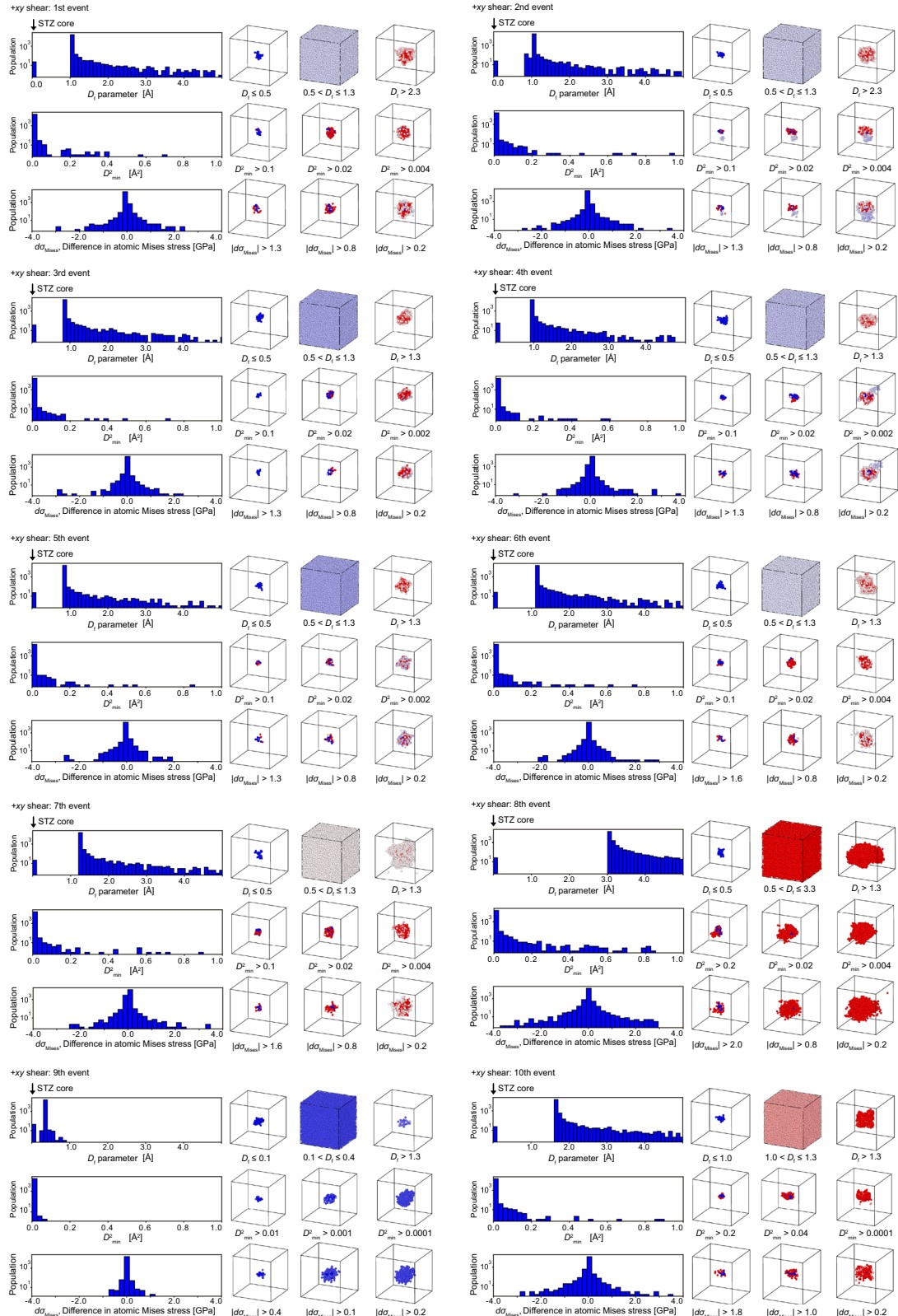

**Fig. 7 | Frozen atom analysis results for ten events in the +*xy* shear direction.** These results follow the same format as Fig. 1 in the main text. Details of the descriptors and visualization methods are described in the caption of Fig. 1.

events under −*xy* and +*xy* shear as examples, illustrating results similar to those in Fig. 1. Note that the center of the gravity of the atomic groups under cooperative motion is centered in these cells. Consistent with the results in the main text, the $D_f$ parameter is obtained following the procedure described in Fig. 1. Additionally, changes in $D_{min}^2$ and von Mises stress are calculated by determining atomic displacements or differences in atomic von Mises stress before and after relaxation at the strain step where the event occurs. In all cases, the $D_f$ parameter histogram successfully isolates deformation units that exhibit cooperative motion. Similar results were obtained for the other 40 cases not

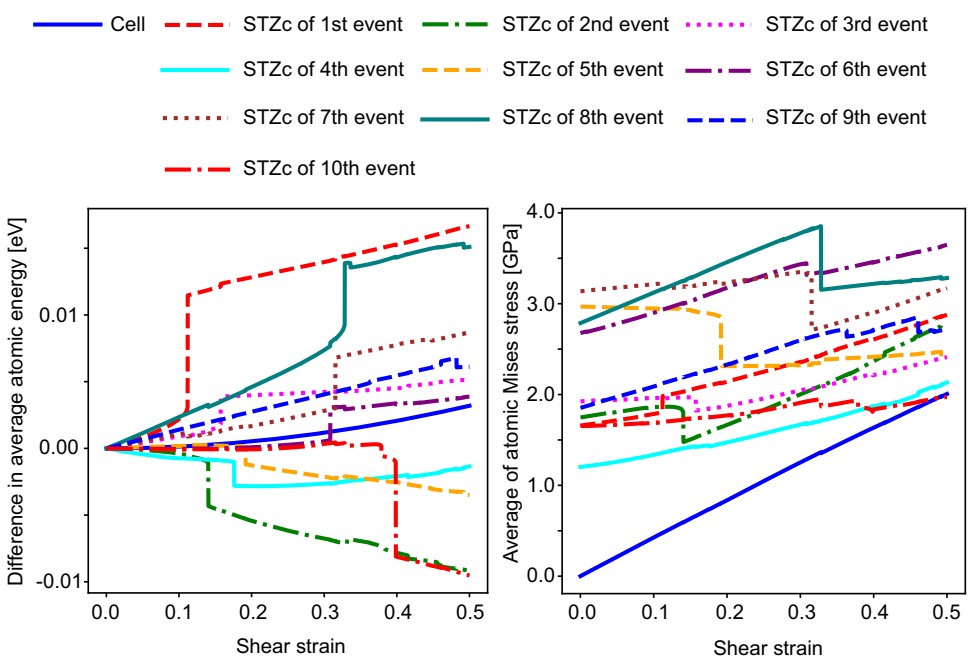

**Fig. 8 | Strain dependent energy and stress histories for each shear transformation zone core.** This figure compares the evolution of average atomic energy (left) and stress state (right) for each shear transformation zone core (STZc) in the −*xy* shear case. The stress state is represented by the von Mises stress, calculated from the average stress components within each STZ core. System-averaged values are also shown for reference, highlighting the deviation of the individual STZ core from the overall material response.

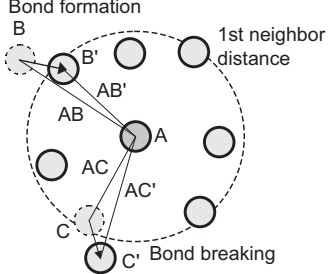

**Fig. 9 | Modification of local atomic connectivity due to bond formation and breaking.** A schematic illustration shows local configurational excitation around a central atom, showing bond formation and bond breaking events within the first-neighbor distance. The dashed circle indicates the cutoff for defining nearest neighbors, while the arrows illustrate changes in atomic connectivity associated with the formation of a new bond and the breaking of an existing bond.

shown here. For most cases, the histogram of the $D_f$ parameter takes the form shown in Fig. 1c: the STZ core appears at $D_f = 0$, a gap opens clearly, and the peak of the $D_f$ parameter emerges. Some cases exhibit different histogram shapes (for example, the first and second events under −*xy* shear), but these correspond to cascades involving the excitation of other STZ cores, as illustrated in Fig. 4. Details are discussed in the next section. Uniquely defining STZ cores enables the calculation of their local physical properties. Figure 8 exemplifies the evolution of average atomic energy and von Mises stress within each STZ core during the AQS process. Atomic energy is computed using the EAM potential, while von Mises stress is derived from the average stress components within each STZ core. The response of individual STZ cores to shear strain at the system level varies significantly, exhibiting distinct behaviors across different cores. Furthermore, the activation of one STZ core strongly influences the properties of others. Even in regions that appear to undergo elastic deformation between stress-drop events, local STZ core responses deviate considerably from the averaged behavior inferred from system-wide values. These findings demonstrate that physical quantities within STZ cores exhibit inherently nonlinear and transient characteristics.

## Local configurational excitation

Local configurational excitation (LCE) represents the reconfiguration of atomic bonding networks, as illustrated in Fig. 9, and has been discussed in the literature as a mechanism that determines the macroscopic viscosity of high-temperature liquids on an atomic level[32]. When LCE occurs—signified by the breaking and forming of atomic bonds—the balance of internal forces changes, leading to alterations in atomic-level stress. Additionally, atoms displace to establish a new equilibrium. Thus, LCE is the phenomenon itself, driving the observed changes. Associated stress relaxation events and atomic displacements are not separate processes but are direct consequences of LCE.

In this study, to examine the relationship between cooperative motion identified by the $D_f$ parameter and LCE, we analyzed the extent to which atoms undergoing LCE were included in groups exhibiting cooperative motion. The results are shown in Fig. 4a of the main text. Determining atomic bonding network reconfiguration is challenging in molecular dynamics; however, we assessed it by tracking atoms entering or leaving the first neighbor shell, following the method outlined in the literature[3] (see Fig. 9). Specifically, for configurations before structural relaxation in stress drop events, we first determined bond presence—whether atoms were in the first neighbor shell—using the pair density function. In this study, Zr–Zr, Cu–Cu, and Zr–Cu bonds were considered part of the first neighbor shell if their interatomic distances were within 4.35, 3.05, and 3.75 Å, respectively. Bond breaking was defined as an atom moving outside the first neighbor shell with an interatomic distance increase beyond a set threshold after relaxation, whereas bond formation was defined as an atom moving into the first neighbor shell with a decrease in interatomic distance beyond the same threshold. The threshold was set at 0.075 Å for this study. We evaluated the total bond breakings and formations per atom across 60 events (totaling 960,000 samples) and generated the histogram shown in Fig. 4.

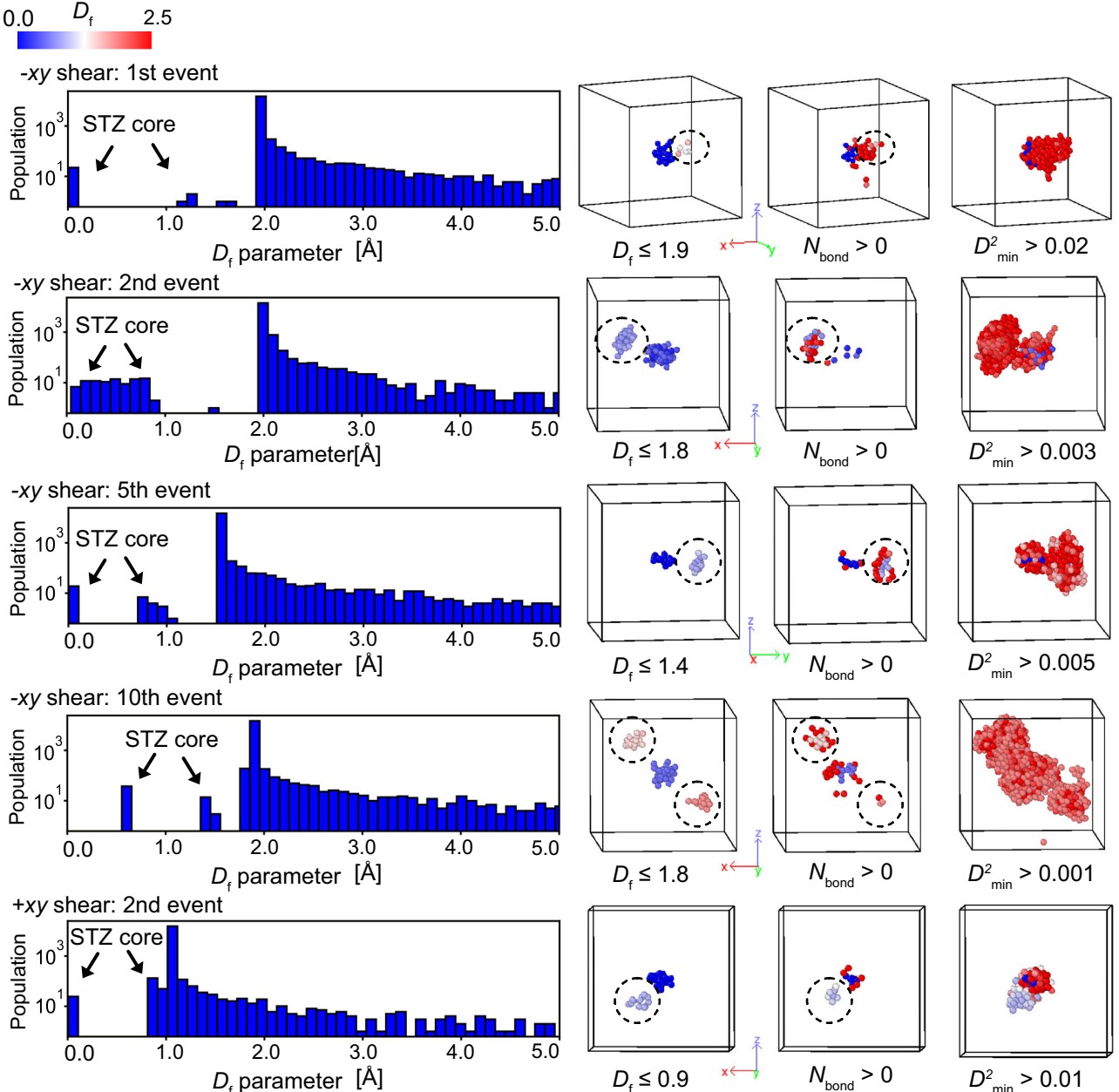

**Fig. 10 | Logarithmic histograms of the $D_f$ parameter for cascade events and corresponding atomic configurations.** In the histograms, arrows indicate the positions of shear transformation zone (STZ) cores, including those that trigger or are triggered by other STZ cores. The atomic configurations highlight STZ core atoms, atoms associated with local configurational excitations (LCEs), and atoms with large non-affine displacements. Dashed circles denote additional STZ cores identified by adjusting the $D_f$ parameter threshold.

LCE analysis is useful for capturing cascade events. Among the 60 stress-drop events examined in this study, five representative cases in which cascades were likely to have occurred are shown in Fig. 10, together with the histograms of the $D_f$ parameter and the atomic structures of STZ cores, atoms involved in LCE, and atoms exhibiting large non-affine displacements. Cascade-like events were identified in 8 of the 60 cases, and the results displayed in the figure encompass all representative patterns observed. Since LCE analysis and non-affine deformation highlight atomic groups distinct from the STZ cores, these observations suggest that cascades are driven by the STZ cores.

Across these cases, four characteristic histogram patterns were identified. The most common was the canonical pattern, in which the STZ core was located at $D_f = 0$, a clear gap opened, and a peak emerged corresponding to the majority of non-core atoms (Fig. 1c). A second pattern, often associated with cascade events, exhibited a small group of atoms appearing before this peak, indicating the involvement of subsequently activated STZ cores (e.g., the first and fifth events under −xy shear and the second event under +xy shear, as exemplified in Fig. 10). In another case, the second event under −xy shear (Fig. 10), the initial STZ core and the subsequently triggered one could not be clearly separated, although such an occurrence was found only once among the 60 events. By contrast, in the 10th event under −xy shear (Fig. 10), the STZ core shifted away from $D_f = 0$, corresponding to a situation in which a single STZ core induced two subsequent STZ cores. These diverse histogram shapes demonstrate multiple modes of interaction among STZ cores, reflecting the complexity of cascade formation mechanisms and highlighting an intriguing direction for future research.

By design, frozen-atom analysis can capture only the initial STZ core, and therefore, the secondary atomic groups observed in some histograms cannot be directly detected by this method. Nevertheless, as shown above, their presence suggests the existence of additional STZ cores whose activation is suppressed when the motion of their constituent atoms is frozen. Taken together, these observations support the following scenario for cascade formation: an STZ core induces non-affine deformation via the LCE it contains, which subsequently triggers another STZ core. This chain reaction, with each step serving as an elementary process, ultimately results in an avalanche that drives large-scale plastic deformation.

## Data availability

The data underlying the figures, including atomic structure data and associated derived quantities, have been deposited in Zenodo with DOI 10.5281/zenodo.17958994[58]. The complete set of atomic configurations exceeds 1 TB and cannot be deposited in a public repository due to size limitations. These data are preserved by the authors and may be accessed for non-commercial academic research by contacting the corresponding author (shiihara@toyota-ti.ac.jp). Requests will receive a response within 2 weeks, and the data will remain available for at least 5 years after publication.

## Code availability

The implementations of the frozen-atom analysis are available in a Code Ocean capsule at DOI 10.24433/CO.2573501.v1[59].

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

## Acknowledgements

This work was supported by Grant-in-Aid for Transformative Research Areas B, "Rheology of disordered structures: establishing Anankeon dynamics", JSPS KAKENHI Grant Numbers 22B206, 22H05041, 22H05042, 22H05040, and 23K28105. T.E. was supported by the U.S. Department of Energy, Office of Science, Basic Energy Sciences, Materials Sciences and Engineering Division.

## Author contributions

Y.S. and T.I. conceived the research. Y.S. developed the modeling framework. Y.S. and T.I. performed the simulations and conducted the data analysis. Y.S. and T.E. developed further interpretation of the data and wrote the manuscript. N.A. and Y.T. provided advice on the physics of the system. All the authors discussed the results and contributed to the manuscript.

## Competing interests

The authors declare no competing interests.
