## [Transparent Peer Review file · Nature Communications]

Cooperative atomic motion during shear deformation in metallic glass

Corresponding Author: Professor Yoshinori Shiihara

Version 0:

Reviewer comments:

Reviewer #1

(Remarks to the Author)

In the submitted manuscript, using numerical simulations of a model metallic glass (CuZr alloy) the authors propose a method for identifying the core of a plastic event occurring by employing a pinning construction. The analysis is done in the athermal quasistatic limit, wherein a plastic event is initially identified and then one reverts back to the state prior to energy minimization and pins individual atoms one by one, and thereafter monitor whether this intervention inhibits the relaxation process by measuring a quantity D_f . $P(D_f)$ exhibits a distinct gap, which is particularly interesting as it suggests the existence of a subpopulation of particles whose pinning significantly suppresses relaxation. This behavior is used to define the core of the shear transformation zone (STZ).

1) The approach is interesting. One should note that there are prior studies on the effect of pinning on mechanical response (Bhowmik et al 123, 185501 2019) where it was shown how plastic activity can be influenced by pinning. In the AQS limit, it is possible to know, using the Hessian approach, which eigenmode will lead to failure. It is intriguing that pinning a particle (within the eigenmode) that is predicted to largely participate in the failure can completely suppress the process. This needs to be properly analyzed. While this study gives us a glimpse of what is possible, I still feel more analysis across diverse models is needed to justify the efficacy of this construction.

2) The authors study $N=2000$ particles. It is not clear what is the lateral size of the simulation box. For the EAM potential, we know that there are long range effects. Hence, one needs to study bigger system sizes, I feel. Also different cooling histories need to be studied, to understand the efficiency of the construction.

3) Also, I would prefer to study a 2d system, even if it is a LJ model, to better visualize the effect of pinning on the eigenmode and compare with the full relaxation (i.e. without pinning).

4) In the usual scenario, once the occurrence of plastic event is identified in an AQS scenario, the distribution of displacements during the event would have the core in the tail, from where one can already identify the participants. That seems to me a quicker analysis than this pinning construction where one has to go through each and every particle, do the relaxation step and then identify the core. This is $O(N)$ process, which for a large enough system size would be time consuming. Is this cost beneficial enough? I can see the usefulness in the case of large cascades to understand where the core was, but there are now other methods to follow the cascade process and identify individual events.

Considering all these, I do not feel that the work presented in this manuscript can be considered for publication in Nature Communications.

Some other feedback -

(i) The manuscript is poorly written in several sections, especially in the discussion of figures. For instance, the content of Fig. 1d,e is not clearly explained in the main text; readers are forced to refer to the Methods section for clarification.

(ii) The description of the amorphous state preparation is sketchy. Without an estimate of the glass transition temperature, it is not clear what a temperature of 2000 K implies. Was the system equilibrated at 2000 K prior to quenching? After cooling, does the kinetic energy effectively reach zero? While this can be inserted in LAMMPS input scripts, we need the the

transient data to clarify what was actually achieved after the cooling. Furthermore, the statement "the atomic structure and cell were relaxed structurally" is vague and needs more precise explanation. Important parameters such as system density and lateral box size are missing—these are critical for reproducibility.

(iii) In the AQS protocol, what is the magnitude of the shear strain used? Were smaller strain increments tested to examine the robustness of the identified plastic events and the proposed pinning-based method?

(iv) The cartoon in Fig.1b is misleading. In reality, we usually study 2d or 3d systems, where pinning of a single point still allows for relaxation, since other pathways are available.

(v) There is already a "frozen atom" construction in the literature (ideated by P. Sollich; see Barbot et al Physical Review E 97, 033001 (2018)), which does the complementary step of the freezing everyone except a sub-population within a specific zone. Such a construction also allows one to identify the first plastic event using an appropriate size of the non-frozen zone. In fact, Barbot et al identify the first few events via their construction which shows that events are not repeated at the same place -- so, we already have such examples.

(vi) In the plot for $P(D_f)$, it would be useful for having x-axis in log-scale to know the overall displacements during the pinning construction.

(Remarks on code availability)

Reviewer #2

(Remarks to the Author)

This paper is concerned with the physical mechanisms underlying shear stress relaxation in metallic glasses subjected to shear strain. This is a topic of practical interest for the design of stronger materials, as well as of fundamental scientific interest as it connects to ideas in complex systems undergoing avalanche-like behavior.

The authors start their work with a well-established computer simulation protocol using the LAMMPS code and a well-established interaction potential for copper-zirconium metallic glass.

The novelty of the paper lies in the specific diagnostic employed to detect atoms that are part of the so-called Shear Transformation Zone (STZ) for specific stress relaxation events. Previous methods to identify STZ atoms have proven unreliable, with results often ambiguous and sensitive to the choice of various threshold values in the analysis. In contrast, the conceptually simple "frozen atom analysis" developed in this work provides a very clear distinction between STZ and non-STZ-atoms. Importantly, the identification of STZ atoms is unambiguous, with the appropriate choice of threshold emerging from the analysis.

I predict that this "frozen atom analysis" will be quickly adopted by other groups and will lead to better understanding of many stress relaxation simulations in glassy systems, and perhaps extending to non-glassy systems.

The paper is exceedingly well written and the presentation of results is clear and compelling. I see no real weaknesses.

I have one suggestion for future work, probably outside the scope of this paper. Both the simulation protocol and the analysis are thermal and quasi-static. In contrast, real glasses always have some small amount of thermal motion. I wonder to what extent the STZ is affected by thermal noise. This could be tested by introducing random displacements of a given average amplitude to all the atoms or just the frozen atom atom prior to the minimization step.

(Remarks on code availability)

Reviewer #3

(Remarks to the Author)

Review report on Manuscript#: NCOMMS-25-25276-T

This manuscript deals with a long-standing issue about the nature of plastic deformation in amorphous solids with a new computational methodology. In order to do this, the authors propose a novel analysis method – the frozen atom analysis -- based on simulations of the athermal quasistatic shear in a CuZr metallic glass sample. This method is based on analysis on the frozen atoms after affine shear back to a state just before the plastic event. The logic is natural, if the frozen atom blocks formation of the shear transformation zone (STZ) upon proceeding shear deformation, it is categorized into the STZ cores; otherwise, it is glassy matrix. With this method it is allowed to recognize two critical steps of deformation, the shear transformation event and the following cascade deformation with STZ as a trigger of activation. The main finding is that the size of STZ core is of about 40 atoms. The STZs are highly stochastic and there is no clear structural feature in them. The STZs deformation acts as triggers where atoms move collectively due to elastic constraints, which induces following large scale deformation cascade that drives strain avalanches as many studied has revealed. In principle, this is a solid study with broad interest in the community of materials science and physics. The proposal method is new, and the conclusion drawn is

impactful in physics. However, there are still a couple of concerns about novelty in science and robustness of conclusions made with the current data at hand. It is suggested to conduct extra studies to enhance reliability of the study and generalize the conclusion to all amorphous solids in terms of plastic deformation mechanism. The suggestions about revisions are listed below.

It is true that the STZs are highly stochastic and can happen everywhere, however their occurring probability is highly dependent on the local environment of atoms. After sufficient spatial and temporal analysis, there is possibility to reveal a sort of structure-property relationship for amorphous solids, see the work by Tong and Tanaka (<https://journals.aps.org/prx/abstract/10.1103/PhysRevX.8.011041>). The utility of 'structure' (e.g., Voronoi cell) in property of glasses has also been quantitatively estimated in both metallic glass and Kob-Anderson model glassy. The indication there is the same to the present conclusion that local structure is irrelevant (Wei et al, J. Chem. Phys. 150, 114502 (2019)). However, there exist a physically meaningful length scale for the correlation between 'defects' and STZs, which is of roughly sub-nano-meter scale (Ref [41] Wei et al., Physical Review B 99, 014115 (2019)). That means statistics is always necessitated when discussing the structural trigger of STZs in amorphous materials from a physical perspective.

Is there a coarse-graining size in computation of the atomic shear modulus? Or instead, the modulus is just estimated by the Voronoi volume a specific atom? If the modulus is defined at atomic scale, it is not important at all, as the authors have pointed out that there is no correlation between STZ and structural feature. However, if the atomic-scale modulus is spatially averaged to some extent, it is quite possible that the peak positions of STZc and non-STZc will be shifted in Fig. 2c. In other words, the referee doubts on the strong conclusion made without careful statistical analysis to some proper space range. As a result, the following statements are overclaims.

"This indicates that STZ cores are neither the "soft spots" as traditionally perceived nor "hard spots"; in other words, they do not exhibit any distinct characteristics in terms of hardness."

"These results suggest that the atoms in STZ cores are not in special environments that could be identified as defects".

"...the formation of STZ cores is a highly stochastic and transient process: unlike conventional deformation units such as dislocations, they do not possess distinct structural features and can form anywhere."

The conclusions are too strong with the limited discussion at hand. It is possible that other method like the topological analysis can provide some hint after taking count of some higher dimensional structural information.

It is claimed that the size of STZ core includes about 40 atoms. However, the referee argues that this size should be strongly affected by the thermodynamic state of the glass model. Recently, Ding et al. (<https://doi.org/10.1073/pnas.2213941119>) has prepared ultrastable glasses with swap Monte Carlo, in which it is found that the size of STZs is about 10 atoms. This is much smaller than claim of several decades here. The referee does not mean that 40 is incorrect, but this quantity is strongly dependent on the structural and energy status of a glass sample. In philosophy, it is again that the nature of STZ is relevant to structure in a broader sense.

The technique of frozen analysis is fine, but the novelty of this method is somehow question. In the literature, Ning Xu (<https://doi.org/10.1073/pnas.2304974120>) has proposed a concept of "stabilizing bonds" that is quite related to instability of amorphous solids, which is quite similar to the frozen atom notion here. It is informative if the author could discuss them together.

What is the meaning of 'N' is equation (1). It is not clear whether it is the number of atoms in the simulation box, or the STZ core region.

The abbreviation "local configurational excitations (LCEs)" has been defined a couple of times.

It is not clear how large is the strain step used in the athermal quasistatic shear, which might slightly affect the definition of STZ core.

(Remarks on code availability)

Version 1:

Reviewer comments:

Reviewer #2

(Remarks to the Author)

I am satisfied with the response to my comments and that of the other reviewers. I recommend publication without further review.

(Remarks on code availability)

Reviewer #3

(Remarks to the Author)
2nd review report

As I stated in my previous report, the idea of "frozen-atom analysis" to identify STZs is both simple and powerful. The proposal carries sufficient novelty to merit publication in Nature Communications. In this revision, the authors have carefully addressed all reviewers' suggestions by employing a larger simulation box, a smaller strain step, and additional cases of STZ analysis. The conclusions remain robust, and all necessary details have been clearly explained and well presented in the manuscript. I consider the work publishable in its current form.

(Remarks on code availability)

I don't have enough time to go through the details of the code about frozen atoms. The results obtained from the code seems physically sound and correct.

Response to reviewers

We sincerely thank the reviewers for their careful reading of our manuscript and for the constructive comments that have greatly improved our work. In response, we have undertaken a substantial revision of the paper. Specifically, we have extended the molecular dynamics calculations from the original 2,000-atom system to a 16,000-atom system, and all simulation results have been recomputed accordingly. In addition, the number of deformation events analyzed has been nearly doubled. Importantly, the core novelty of our approach is preserved in this extended analysis, while the results have become physically more robust and convincing. Thus, the reviewers' suggestions have significantly strengthened the study.

We provide a point-by-point response below. The submission consists of three components: the marked manuscript (differences from the previous version), the clean manuscript, and the Supplementary Materials. **Blue highlighting indicates revisions/additions in the marked manuscript**; line numbers cited correspond to that marked version.

Reviewer #1 (Remarks to the Author):

In the submitted manuscript, using numerical simulations of a model metallic glass (CuZr alloy) the authors propose a method for identifying the core of a plastic event occurring by employing a pinning construction. The analysis is done in the athermal quasistatic limit, wherein a plastic event is initially identified and then one reverts back to the state prior to energy minimization and pins individual atoms one by one, and thereafter monitor whether this intervention inhibits the relaxation process by measuring a quantity Df . $P(Df)$ exhibits a distinct gap, which is particularly interesting as it suggests the existence of a subpopulation of particles whose pinning significantly suppresses relaxation. This behavior is used to define the core of the shear transformation zone (STZ).

Q1-1:

1) The approach is interesting. One should note that there are prior studies on the effect of pinning on mechanical response (Bhowmik et al 123, 185501 2019) where it was shown how plastic activity can be influenced by pinning. In the AQS limit, it is possible to know, using the Hessian approach, which eigenmode will lead to failure. It is intriguing that pinning a particle (within the eigenmode) that is predicted to largely participate in the failure can completely suppress the process. This needs to be properly analyzed. While this study gives us a glimpse of what is possible, I still feel more analysis across diverse models is needed to justify the efficacy of this construction.

A1-1:

We thank the reviewer for the positive assessment of our approach. We agree that pinning is known to be a powerful approach to probe the nature of mechanical deformation and the glass transition. While related in spirit, our frozen-atom analysis is fundamentally different from the well-known random pinning: we freeze only the motion of a single atom just before each stress drop and apply this procedure independently to all atoms, which makes our approach completely new and promising as a method to identify the STZ cores, the atomic groups in cooperative motion. As the reviewer suggested, applying our frozen-atom analysis to participating particles identified from the Hessian matrix might indeed yield similar results and could stimulate further studies along this line. We believe this possibility highlights the novelty of our approach. At this stage, our emphasis is on detecting STZ cores and clarifying their statistical properties with our new method, while we regard its relation to eigenmodes as a promising direction for future studies.

Instead, we have substantially strengthened the present study by enlarging the system size from 2,000 to 16,000 atoms and by recomputing all simulation results and repeating the analysis to ensure much greater robustness. The details are provided in A1-2 and the subsequent responses. We hope that these extensive revisions adequately address the reviewer's main concerns.

Q1-2:

The authors study $N=2000$ particles. It is not clear what is the lateral size of the simulation box. For the EAM potential, we know that there are long range effects. Hence, one needs to study bigger system sizes, I feel. Also different cooling histories need to be studied, to understand the efficiency of the construction.

A1-2:

We thank the reviewer for this valuable comment. The lateral size of the simulation box with 2,000 atoms is about 33 Å, and the interaction range of EAM potential used is 6.5 Å. Therefore, we did not expect the 2,000-atom system to cause serious computational artifacts, and indeed similar system sizes are commonly used in studies employing the activation–relaxation technique (Fan 2014, Fan 2015). Nevertheless, we recognize that the displacement field generated by the excitation of an STZ can extend over long distances, and the 2,000-atom simulation cell might not have been sufficiently large. In the present revision, we have enlarged the system volume by a factor of eight, corresponding to 16,000 atoms, which is comparable to those employed in earlier STZ studies (e.g., Ding 2014; Zhang 2022), and therefore provides a more robust basis for the present analysis. The changes are summarized as follows:

- **System size expansion:** The MD model was enlarged from 2,000 atoms to 16,000 atoms, and all AQS simulations were recomputed (Methods, “Simulation Setup” in Supplementary materials).
- **Increased number of events:** The analyzed stress-drop events were increased from 36 to 60, and all related figures and tables were updated (Main Text Figs. 1–8, 10; Supplementary Figs. S1-S2, Table S1).
- **Strengthened statistics** (Voronoi / LCE analyses): The statistics of Voronoi features, atomic elasticity,

and bond formation/breaking were expanded from 72,000 samples to 960,000 samples (60 events \times 16,000 atoms), and histograms were reconstructed (Fig. 2, Fig. 4a).

- **Cascade analysis extended:** Out of 60 events, 8 exhibited cascade-like behavior, and three characteristic histogram patterns were summarized with representative examples (Marked manuscript: Lines 449-484, Fig. 10).

Despite the significant expansion of system size and statistical sampling, the core findings regarding the STZ cores remain unchanged, confirming the robustness of our conclusions.

Having considered the reviewer's comment, in the 2,000-atom simulations, we realized that the displacement field of a single STZ core was large enough to influence the entire cell (as shown in Fig. 8 of the previous version). This indicates that in small cells, STZ cores may interact with their own periodic images. In contrast, with 16,000 atoms such finite-size artifacts are no longer observed. This is demonstrated in the revised Fig. 8, where only a limited number of other STZ cores are affected by the excitation of a given STZ core. As a result, the results have become physically more robust and convincing.

The reviewer also requested simulations at different cooling rates. We acknowledge that this is indeed an interesting and important question. We consider this to be a separate research topic on how the stability of glass affects the properties of STZ cores, as previous work has demonstrated that cooling rate can significantly modify the size and population of STZs (Zhang 2022). Such an investigation will therefore be better addressed in future work.

Reference:

Fan, Y., Iwashita, T., & Egami, T. (2014). How thermally activated deformation starts in metallic glass. *Nature Communications*, 5, 5083.

Fan, Y., Iwashita, T., & Egami, T. (2015). Crossover from localized to cascade relaxations in metallic glasses. *Physical Review Letters*, 115(4), 045501.

Ding, J., Patinet, S., Falk, M. L., Cheng, Y., & Ma, E. (2014). Soft spots and their structural signature in a metallic glass. *Proceedings of the National Academy of Sciences*, 111(39), 14052-14056.

Zhang, Z., Ding, J., & Ma, E. (2022). Shear transformations in metallic glasses without excessive and predefinable defects. *Proceedings of the National Academy of Sciences*, 119(48), e2213941119.

Q1-3:

Also, I would prefer to study a 2d system, even if it is a LJ model, to better visualize the effect of pinning on the eigenmode and compare with the full relaxation (i.e. without pinning).

A1-3:

We appreciate the reviewer's suggestion of studying a two-dimensional (2D) system and acknowledge its advantage for easier visualization. At the same time, we believe this would open up a very different line of investigation. In 2D, the elastic energy of dipolar stress fields diverges, which leads to screening effects and

potentially inducing spurious atomic behavior (Egami 2011). As a result, the fundamental physics is qualitatively different from that of 3D metallic glasses. More recent simulation work has also demonstrated that the structural ordering mechanisms in 2D metallic glasses differ essentially from those in 3D, for example, crystal-like hexagonal order dominates in 2D, whereas icosahedral order governs the 3D case (Hu 2017). These findings highlight that 2D and 3D systems raise distinct sets of questions. For this reason, we consider that pursuing the 2D case, while interesting, would be beyond the scope of the present study.

Egami, T. (2011). Atomic level stresses. *Progress in Materials Science*, 56(6), 637-653.

Hu, Y. C., Tanaka, H., & Wang, W. H. (2017). Impact of spatial dimension on structural ordering in metallic glass. *Physical Review E*, 96(2), 022613.

Q1-4:

In the usual scenario, once the occurrence of plastic event is identified in an AQS scenario, the distribution of displacements during the event would have the core in the tail, from where one can already identify the participants. That seems to me a quicker analysis than this pinning construction where one has to go through each and every particle, do the relaxation step and then identify the core. This is $O(N)$ process, which for a large enough system size would be time consuming. Is this cost beneficial enough? I can see the usefulness in the case of large cascades to understand where the core was, but there are now other methods to follow the cascade process and identify individual events.

A1-4:

Our central claim is that an STZ should be regarded as consisting of two parts: the indispensable atomic group that acts cooperatively to trigger the excitation (the STZ core) and the surrounding atoms whose displacements are induced by it. We appreciate the reviewer's suggestion that examining the displacement distribution during a plastic event may provide a quicker way to identify the active atoms. We would like to point out that this approach is essentially equivalent to analyzing the D^2_{\min} field, since D^2_{\min} effectively captures the displacement distribution. As shown in Fig. 1(d), D^2_{\min} indeed highlights the spatial zone of activity, yet it provides only an approximate indication of the core and does not allow us to identify the indispensable atoms that constitute the core itself. By contrast, the frozen-atom analysis enables us to extract the STZ core directly and without ambiguity. As noted in Q1-5, this distinction was not sufficiently emphasized in the previous version, and we have now added explicit statements in the revised manuscript as explained in A1-5.

On the other hand, we would like to thank the reviewer for drawing our attention to the important issue of computational cost. Inspired by the reviewer's comment, we recognized that using atomic displacement as an initial screening step made it possible to greatly reduce the computational cost of the frozen-atom analysis. By implementing this strategy, we were able to reduce the computational complexity of the frozen-atom analysis from $O(N^2)$ to $O(N)$. Furthermore, this remains substantially lower than the $O(N^3)$ scaling required for a full eigenmode analysis, providing a significant computational advantage especially for large-scale simulations.

The following explanation has been added to the Supplementary Materials.

Changes in the manuscript:

(Supplementary Materials, Section "Scalability of frozen-atom analysis")

In terms of big-O notation with the number of atoms N , conventional molecular dynamics simulations scale as $O(N)$, whereas performing frozen-atom analysis by freezing every atom scales as $O(N^2)$. To reduce the computational burden, we applied the frozen-atom analysis only to atoms that exhibited a displacement norm above a certain threshold (e.g., 0.01 Å) during the stress-drop event. Using atomic displacement as an initial screening step made it possible to greatly reduce the computational cost of the frozen-atom analysis. While an STZ core contains at most about 100 atoms, the threshold was chosen so that an atomic group roughly ten times larger was included for analysis. An illustrative example is shown in the figure below, where the STZ core is fully contained within the selected group. Test calculations for 20 events under $+xy$ and $-xy$ shear confirmed that the STZ cores identified by this efficient frozen-atom analysis are identical to those obtained by full analysis. Because the size of the STZ core is independent of the total number of atoms in the system, it is evident that freezing all atoms is unnecessary. With this refinement, the frozen-atom analysis can be regarded as an $O(N)$ algorithm, which makes the method scalable to large systems.

Fig. Example of the computational cost reduction in frozen-atom analysis, illustrated for the second event under $+xy$ shear. (a) The STZ core (96 atoms) identified as the group of atoms with D_f . (b) The set of atoms that exhibited displacement norm $|\mathbf{d}|$ greater than 0.01 Å (1204 atoms). (c) Superposition of (a) and (b). As seen in (c), the STZ core can be determined by applying the frozen-atom analysis only to the subset of atoms with relatively large displacements.

Q1-5:

The manuscript is poorly written in several sections, especially in the discussion of figures. For instance, the content of Fig. 1d,e is not clearly explained in the main text; readers are forced to refer to the Methods section for clarification.

A1-5:

We thank the reviewer for pointing this out and apologize for the lack of clarity in the previous version. Particularly, Fig. 1d and 1e were not sufficiently clear in the main text. In the revised manuscript we have

revised the explanations in the main text as follows, so that readers can now understand the contents without having to refer to the Methods section. In addition, brief explanatory sentences were also added for Figs. 2a and 2b to clarify how to interpret the color scale and circle size, respectively.

Changes in the manuscript:

(Marked manuscript: Lines 155-168)

For comparison, Figures 1d and 1e show histograms of conventional indicators: the non-affine displacement measure D^2_{\min} and the change in atomic von Mises stress during relaxation.

The atoms in the tail of the distribution in Figure 1d, namely, those with large non-affine deformation, are often assumed to represent the central region of the STZ. Similarly, in Figure 1e the atoms exhibiting the most pronounced changes in local von Mises stress are also regarded as part of this region. In both cases, the highlighted atoms overlap considerably with those identified by the D^2_{\min} parameter in Fig. 1c. This indicates that, although conventional descriptors capture atoms involved in the event, the size of the atomic group they identify can vary greatly depending on the chosen threshold, which makes their interpretation ambiguous. In contrast, the frozen-atom analysis provides an unambiguous distinction between the indispensable STZ core (the group close to D^2_{\min}) and the surrounding atoms.

(Marked manuscript: Lines 180-184)

Figure 2a illustrates the spatial distribution ... throughout the material. **The color scale shows how often each atom was classified as part of an STZ core.**

(Marked manuscript: Lines 186-190)

Figure 2b plots the features of ... this calculation. **The circle size represents the D_i parameter, where larger circles basically correspond to atoms identified as participating in cooperative motion.**

Q1-6:

The description of the amorphous state preparation is sketchy. Without an estimate of the glass transition temperature, it is not clear what a temperature of 2000 K implies. Was the system equilibrated at 2000 K prior to quenching? After cooling, does the kinetic energy effectively reach zero? While this can be inserted in LAMMPS input scripts, we need the the transient data to clarify what was actually achieved after the cooling. Furthermore, the statement "the atomic structure and cell were relaxed structurally" is vague and needs more precise explanation. Important parameters such as system density and lateral box size are missing—these are critical for reproducibility.

A1-6:

We thank the reviewer for these constructive comments. In the revised Methods section, we have clarified the preparation of the amorphous state in detail as follows. We have also added the missing parameters required for reproducibility, including the system density and the box size. In addition, the transient data of the quenching process have been included in the Supplementary Materials. From the figure below, we find that the glass transition temperature is about 750 K, indicating that 2000 K is sufficiently high to ensure equilibration with a cooling rate of 10^9 K/s.

Changes in the manuscript:

(Marked manuscript: Lines 351-362)

The metallic glass was prepared by melting and quenching a B2 CuZr alloy. A velocity distribution corresponding to 2000 K, which is well above the glass transition temperature of this alloy (approximately 750 K, see Supplementary Fig. S1), was applied to the B2 structure, and the system was directly cooled to 0 K at a rate of 10^9 K/s in an NVT ensemble using the Nosé-Hoover thermostat[44,45] without an intermediate annealing period. After cooling, the atomic structure and simulation cell were relaxed using the conjugate gradient method until all stress components converged to below 0.002 MPa. The final simulation box had a characteristic length of approximately 65.4 Å and was nearly cubic but with slight tilts in the cell vectors, as reported in the Supplementary Materials.

(Supplementary Materials, Section "Preparation of the model glass structure")

As described in the Methods section, in this study a velocity distribution corresponding to 2000 K was applied to the B2 structure of $\text{Cu}_{50}\text{Zr}_{50}$, and the system was directly cooled to 0 K at a rate of 10^9 K/s without an intermediate annealing stage. The transient data of the average total energy per atom and the average potential energy per atom obtained during this process are shown in Supplementary Fig. S2. The potential energy exhibited a clear change in slope at around 750 K, from which the glass transition temperature T_g of this model glass was estimated to be approximately 750 K. Since the initial temperature of 2000 K is well above this value, the system can be regarded as having been quenched from a fully equilibrated liquid state. At 0 K, the total and potential energies coincide, indicating that the kinetic energy has vanished (less than 2 meV/atom). After cooling, the atomic structure and simulation cell were relaxed using the conjugate gradient method until all stress components converged to below 0.002 MPa. The final cell configuration of the $\text{Cu}_{50}\text{Zr}_{50}$ glass after quenching and relaxation is provided in Table S1 to ensure reproducibility. The simulation box has an edge length of approximately 65.4 Å and is nearly cubic, but with small tilt factors, indicating a slightly triclinic geometry. The system density was 7.38 g/cm³ after quenching and relaxation. The system density and T_g obtained here are consistent with those reported in Ref. [4] where the same Embedded-Atom Method (EAM) potential was employed, showing no significant discrepancy.

Figure S2. Preparation of the model glass: evolution of per-atom total and potential energies during rapid quenching. The glass transition temperature, T_g is estimated to be ~ 750 K from the change in slope of the potential-energy curve.

Table S1. Final cell parameters of the $\text{Cu}_{50}\text{Zr}_{50}$ glass after quenching and relaxation

Vector	x (Å)	y (Å)	z (Å)
a	65.4106	0.0098	-0.0314
b	0.0060	65.2043	0.0473
c	0	0	65.2881

Q1-7:

In the AQS protocol, what is the magnitude of the shear strain used? Were smaller strain increments tested to examine the robustness of the identified plastic events and the proposed pinning-based method?

A1-7:

We thank the reviewer for raising this important question. We agree that the choice of strain increment in the AQS protocol is critical to avoid multiple plastic events at the same time. In the previous version of the manuscript, we used a strain increment of 5×10^{-5} per step. In the revised work, following the practice in the relevant literature (Zhang2022), we have reduced the increment to 1×10^{-5} per step. This clarification has been explicitly added to the Methods section.

Changes in the manuscript:

(Marked manuscript: Line 367-369)

This procedure was repeated 15,000 times to induce deformation until the engineering shear strain reached

15%, with a strain increment of 10^{-5} per step.

Reference:

Zhang, Z., Ding, J., & Ma, E. (2022). Shear transformations in metallic glasses without excessive and predefinable defects. *Proceedings of the National Academy of Sciences*, 119(48), e2213941119.

Q1-8: *The cartoon in Fig.1b is misleading. In reality, we usually study 2d or 3d systems, where pinning of a single point still allows for relaxation, since other pathways are available.*

A1-8:

We thank the reviewer for pointing out the possible misunderstanding caused by the schematic cartoon in Fig. 1b. We agree that the illustration is an oversimplification, since a three-dimensional glass indeed has many more degrees of freedom. The intent of the cartoon is only to convey the conceptual basis of our method: freezing the motion of a single atom can disrupt the cooperative motion of the entire group, thereby highlighting the interdependence of atoms within the STZ core. To avoid confusion, we have revised the text to explicitly state this limitation:

Changes in the manuscript:

(Marked manuscript: Line 104-108)

“Figure 1b schematically illustrates that freezing a single atom disrupts the cooperative motion of the entire group, emphasizing their interdependence, although this representation is oversimplified because a three-dimensional glass has much higher degrees of freedom.”

We hope this clarification makes clear that the figure is not meant as a literal description of the system but as a conceptual aid. For this reason, we would prefer to retain the cartoon. Nevertheless, we remain open to removing it should the reviewer still find it inappropriate.

Q1-9:

There is already a "frozen atom" construction in the literature (ideated by P. Sollich; see Barbot et al Physical Review E 97, 033001 (2018)), which does the complementary step of the freezing everyone except a sub-population within a specific zone. Such a construction also allows one to identify the first plastic event using an appropriate size of the non-frozen zone. In fact, Barbot et al identify the first few events via their construction which shows that events are not repeated at the same place - - so, we already have such examples.

A1-9:

Because of the multidimensional nature of glassy response, many approaches have been proposed to reveal the essential physics by constraining part of the motion. Our frozen-atom analysis is distinct in that it directly identifies the number of atoms that cooperatively trigger a stress drop, rather than serving as a method to detect

the plastic events themselves. Nevertheless, it is important to clarify how our method should be positioned within this broader context, and we are grateful to the reviewers for drawing our attention to this point. In response to their comments, we have added a discussion in the Supplementary Materials introducing the related studies noted by Reviewer #1 (Bhowmik 2019, Barbot 2018) and Reviewer #3 (Xu 2023), and explicitly describing how our method differs from them, thereby underscoring the novelty of the present work.

Changes in the manuscript:

(Marked manuscript: Lines 120-124)

This method appears similar to the artificial manipulations such as atomic pinning approach to probe atomic cooperativity in flow[24, 25] or other related interventions[26, 27], but our approach is quite different in that atomic freezing is used merely to probe the role of each atom in deformation, and it does not affect the flow itself (see Supplementary Materials for details).

(Supplementary Materials, in Section " Frozen-atom analysis and artificial perturbations ")

Before the development of frozen-atom analysis, several studies introduced artificial manipulations to probe the physics of disordered structures. Pinning is one such approach: Berthier, *et al.* constrained the motion of randomly selected atoms under NVT conditions and thereby revealed hidden static order (point-to-set correlations) in glass forming liquids[2]. Bhowmik, *et al.* applied pinning during AQS deformation of a model amorphous solid, showing that frozen atoms suppress avalanche-like plastic events[3]. Barbot, *et al.* instead constrained the surroundings of a local region inside model amorphous solids and imposed affine deformation to forcibly trigger STZs, providing a map of local yielding propensity[1]. Xu, *et al.* proposed to remove the contribution of specific pair bonds from the force constant matrix to identify those whose elimination directly triggers instabilities[5].

Frozen-atom analysis likewise employs artificial manipulation, but differs from pinning method in that it does not suppress deformation indiscriminately throughout the AQS process. Compared with Barbot's method, the distinction lies in whether the constraint is applied inside or outside the STZ, while Xu's approach identifies critical bonds related to Local Configurational Excitation (LCE) rather than cooperative cores. The essential novelty of frozen-atom analysis is its ability to extract cooperative motion directly, a feature not accessible by previous methods. This indicates that frozen-atom analysis and these methods are complementary, and their combination may yield further insights into glass physics.

References:

- Barbot, A., Lerbinger, M., Hernandez-Garcia, A., García-García, R., Falk, M. L., Vandembroucq, D., & Patinet, S. (2018). Local yield stress statistics in model amorphous solids. *Physical Review E*, 97(3), 033001.
- Bhowmik, B. P., Chaudhuri, P., & Karmakar, S. (2019). Effect of pinning on the yielding transition of amorphous solids. *Physical review letters*, 123(18), 185501.
- Xu, D., Zhang, S., Liu, A. J., Nagel, S. R., & Xu, N. (2023). Discontinuous instabilities in disordered solids. *Proceedings of the National Academy of Sciences*, 120(34), e2304974120.

Q1-10:

In the plot for $P(D_f)$, it would be useful for having x-axis in log-scale to know the overall displacements during the pinning construction.

A1-10:

We thank the reviewer for this suggestion. We have tested plotting the distribution $P(D_f)$ with a logarithmic x -axis. In presenting our main claim, namely that the frozen-atom analysis enables a simple separation between the STZ core and the surrounding atoms, we found that the current linear-scale representation is more straightforward and easier for readers to follow. At the same time, we agree that a logarithmic representation may provide additional insights when probing the detailed structure of the STZ core. We therefore regard this as a useful complementary approach and have noted it as a direction for future work.

Reviewer #2 (Remarks to the Author):

This paper is concerned with the physical mechanisms underlying shear stress relaxation in metallic glasses subjected to shear strain. This is a topic of practical interest for the design of stronger materials, as well as of fundamental scientific interest as it connects to ideas in complex systems undergoing avalanche-like behavior.

The authors start their work with a well-established computer simulation protocol using the LAMMPS code and a well-established interaction potential for copper-zirconium metallic glass.

The novelty of the paper lies in the specific diagnostic employed to detect atoms that are part of the so-called Shear Transformation Zone (STZ) for specific stress relaxation events. Previous methods to identify STZ atoms have proven unreliable, with results often ambiguous and sensitive to the choice of various threshold values in the analysis. In contrast, the conceptually simple "frozen-atom analysis" developed in this work provides a very clear distinction between STZ and non-STZ-atoms. Importantly, the identification of STZ atoms is unambiguous, with the appropriate choice of threshold emerging from the analysis.

I predict that this "frozen-atom analysis" will be quickly adopted by other groups and will lead to better understanding of many stress relaxation simulations in glassy systems, and perhaps extending to non-glassy systems.

The paper is exceedingly well written, and the presentation of results is clear and compelling. I see no real weaknesses.

Q2-1:

I have one suggestion for future work, probably outside the scope of this paper. Both the simulation protocol and the analysis are thermal and quasi-static. In contrast, real glasses always have some

small amount of thermal motion. I wonder to what extent the STZ is affected by thermal noise. This could be tested by introducing random displacements of a given average amplitude to all the atoms or just the frozen atom prior to the minimization step.

A2-1:

We sincerely thank the reviewer for the positive evaluation of our work and for highlighting its novelty. As pointed out, we also hope that this method will prove useful to many researchers in materials science. We are furthermore grateful for the constructive suggestion regarding the influence of thermal fluctuations on the stability of STZ cores. We recognize that clarifying the relationship between the characteristics of the STZ core and its stability, as well as exploring how the frozen-atom analysis can be applied to dynamic phenomena, represents an important and intriguing direction for future research. We intend to address these issues in our forthcoming studies.

Reviewer #3 (Remarks to the Author):

This manuscript deals with a long-standing issue about the nature of plastic deformation in amorphous solids with a new computational methodology. In order to do this, the authors propose a novel analysis method – the frozen-atom analysis -- based on simulations of the athermal quasistatic shear in a CuZr metallic glass sample. This method is based on analysis on the frozen atoms after affine shear back to a state just before the plastic event. The logic is natural, if the frozen atom blocks formation of the shear transformation zone (STZ) upon proceeding shear deformation, it is categorized into the STZ cores; otherwise, it is glassy matrix. With this method it is allowed to recognize two critical steps of deformation, the shear transformation event and the following cascade deformation with STZ as a trigger of activation. The main finding is that the size of STZ core is of about 40 atoms. The STZs are highly stochastic and there is no clear structural feature in them. The STZs deformation acts as triggers where atoms move collectively due to elastic constraints, which induces following large scale deformation cascade that drives strain avalanches as many studied has revealed. In principle, this is a solid study with broad interest in the community of materials science and physics. The proposal method is new, and the conclusion drawn is impactful in physics. However, there are still a couples of concerns about novelty in science and robustness of conclusions made with the current data at hand. It is suggested to conduct extra studies to enhance reliability of the study and generalize the conclusion to all amorphous solids in terms of plastic deformation mechanism. The suggestions about revisions are listed below.

Q3-1:

It is true that the STZs are highly stochastic and can happen everywhere, however their occurring probability is highly dependent on the local environment of atoms. After sufficient spatial and temporal analysis, there is possibility to reveal a sort of structure-property relationship for amorphous solids, see the work by Tong and Tanaka (<https://journals.aps.org/prx/abstract/10.1103/>

PhysRevX.8.011041). The utility of ‘structure’ (e.g., Voronoi cell) in property of glasses has also been quantitatively estimated in both metallic glass and Kob-Anderson model glassy. The indication there is the same to the present conclusion that local structure is irrelevant (Wei et al, *J. Chem. Phys.* 150, 114502 (2019)). However, there exist a physically meaningful length scale for the correlation between ‘defects’ and STZs, which is of roughly sub-nano-meter scale (Ref [41] Wei et al., *Physical Review B* 99, 014115 (2019)). That means statistics is always necessitated when discussing the structural trigger of STZs in amorphous materials from a physical perspective.

Is there a coarse-graining size in computation of the atomic shear modulus? Or instead, the modulus is just estimated by the Voronoi volume a specific atom? If the modulus is defined at atomic scale, it is not important at all, as the authors have pointed out that there is no correlation between STZ and structural feature. However, if the atomic-scale modulus is spatially averaged to some extent, it is quite possible that the peak positions of STZc and non-STZc will be shifted in Fig. 2c. In other words, the reviewer doubts on the strong conclusion made without careful statistical analysis to some proper space range. As a result, the following statements are overclaims.

“This indicates that STZ cores are neither the “soft spots” as traditionally perceived nor “hard spots”; in other words, they do not exhibit any distinct characteristics in terms of hardness.”

“These results suggest that the atoms in STZ cores are not in special environments that could be identified as defects”.

“...the formation of STZ cores is a highly stochastic and transient process: unlike conventional deformation units such as dislocations, they do not possess distinct structural features and can form anywhere.”

The conclusions are too strong with the limited discussion at hand. It is possible that other method like the topological analysis can provide some hint after taking count of some higher dimensional structural information.

A3-1:

We thank the reviewer for this valuable comment, which helped us clarify the scope of our study. First, we agree with the reviewer’s opinion that the probability of STZ activation is highly dependent on the local environment of atoms. We understand that the “local environment” in this context refers to the *current configuration* immediately before the event, rather than the *initial configuration* at zero strain. This distinction is crucial, because if no such features are present in the initial configuration, it indicates that STZs are not embedded defects but transient and stochastic excitations, and our definition of the STZ core makes it possible to discuss this point unambiguously for the first time. In this paper we focused on whether features in the initial configuration could provide precursors of STZ cores—in other words, whether STZ cores exist as intrinsic defects in the structure—and addressed this question in Fig. 2.

Regarding the reviewer’s concern about the definition of the atomic shear modulus, we performed additional calculations in direct response to this comment. Specifically, the coarse-graining was carried out for the atomic groups belonging to the identified STZ cores, each consisting of several tens of atoms. Within these groups,

we averaged the shear atomic elasticity, which effectively smooths out atomic-scale noise and yields a meaningful local shear modulus. Even with this coarse-graining, some STZ cores appeared stiffer and others softer than the cell average, with no systematic trend. Combined with the results of Voronoi indices and Voronoi volumes, these new calculations confirmed that STZ cores cannot be identified as defects in the initial configuration. The discussion on these results is now described in the manuscript as shown below.

Although the previous version already stated that our focus is on the features of the initial configuration, this point may not have been conveyed with sufficient clarity. To address this, we have revised and clarified the relevant parts of the manuscript. (In the previous version, we noticed an incorrect description stating that STZ cores were analyzed “just prior to the deformation events,” even though other parts of the manuscript discussed the initial configuration. This descriptive error has been corrected in the revised manuscript [Marked manuscript, Lines 269–272], and we apologize for the confusion caused.)

Changes in the manuscript:

(Marked manuscript: Lines 32-33)

These cores show no clear structural or elastic precursors **in the initial configuration**, challenging the idea that deformation occurs in defective regions.

(Marked manuscript: Lines 176-179)

We studied the atomic-level characteristics of the STZ core atoms **in the zero-strain initial configuration before any shear was applied**, in order to test whether precursors of STZ cores exist.

(Marked manuscript: Lines 186-188)

Figure 2b plots the features of Voronoi polyhedra, specifically the number of faces and the volume **in the initial configuration**, for the 100,000 atoms randomly sampled from 960,000 atoms considered in this calculation.

(Marked manuscript: Lines 197-203)

To further examine the elastic properties, we conducted coarse-grained evaluations of atomic elasticity based on the STZ cores. Even in the undeformed configuration, variability was evident: some STZ cores were stiffer than the cell average, while others were softer. The average shear moduli of the STZ cores were 40.5 ± 3.8 GPa (xy), 40.7 ± 3.1 GPa (yz), and 42.0 ± 3.9 GPa (zx), compared with the overall averages of 41.8, 41.9, and 41.7 GPa, respectively. This indicates that STZ cores **in the initial state** are neither the “soft spots” as traditionally perceived nor “hard spots”; in other words, they do not exhibit any distinct characteristics in terms of elastic constant.

(Marked manuscript: Lines 269-272)

However, the analysis of the STZ core atoms **in the undeformed initial state** did not reveal any distinctive elastic characteristics. STZs are statistically created and disappear after deformation events, as presumed in the STZ theory by Langer[34].

(Marked manuscript: Lines 288-291)

Unlike defects that can be identified from static configurations **before deformation**, STZ cores emerge dynamically through cooperative atomic motion in response to the nonlinear stress and strain behavior of a glassy structure under external stimuli.

Next, we do not dispute the reviewer's view that the structural characteristics of STZ cores may be extractable through topological analyses involving some higher-dimensional structural information. Identifying the features of local structures in the current configuration that lead to instability, and locating similar structures within the disordered system, is important for advancing the understanding of deformation mechanisms in glassy materials. Methods such as graph neural networks might provide a way to detect subtle features that are not readily identifiable by human inspection. Let us emphasize that such approaches can only become meaningful once the boundary of the STZ core has been clearly defined, which highlights the importance of the frozen-atom analysis. We have added this point to the Discussion section. We thank the reviewer for enabling us to emphasize more clearly the advantage of the STZ core concept, namely that it provides an unambiguous definition of the core boundary.

Changes in the manuscript:

(Marked manuscript: Lines 297-306)

STZ cores, or their specific characteristics, may nevertheless be extractable through topological analyses that incorporate higher-dimensional structural information. Here, this refers to describing atoms collectively rather than individually, consistent with previous reports that collective structural diversity rather than specific local motifs governs the dynamics of amorphous alloys[36, 37]. Graph neural networks, which have recently been applied in the context of STZs[38], may capture subtle features not readily identifiable by conventional analysis. Such approaches, however, become meaningful only after the nature of the STZ is identified and the boundary of the STZ core is clearly defined; frozen-atom analysis provides this definition unambiguously by directly identifying the indispensable atoms whose immobilization blocks the cooperative process.

Q3-2:

It is claimed that the size of STZ core includes about 40 atoms. However, the referee argues that this size should be strongly affected by the thermodynamic state of the glass model. Recently, Ding et al. (<https://doi.org/10.1073/pnas.2213941119>) has prepared ultrastable glasses with swap Monte Carlo, in which it is found that the size of STZs is about 10 atoms. This is much smaller than claim of several decades here. The referee does not mean that 40 is incorrect, but this quantity is strongly dependent on the structural and energy status of a glass sample. In philosophy, it is again that the nature of STZ is relevant to structure in a broader sense.

A3-2:

Although the definition of STZ size in our study differs from that used by Zhang, Ding, and Ma (Zhang2022), where atoms with a D_{\min}^2 value above a certain threshold are considered to belong to an STZ, we cannot rule out the possibility that the cooling rate affects the STZ core size. As discussed in A1-4, because the STZ core is embedded within a group of atoms exhibiting a certain magnitude of D_{\min}^2 , if the D_{\min}^2 values become smaller under slower cooling conditions, the STZ core could also become smaller. Therefore, we find the reviewer's concern reasonable, and in accordance with this comment we have removed the statement referring to "40

atoms.” What we wish to emphasize here is that, unlike the other methods compared in Fig. 1, the frozen-atom analysis makes it possible to draw a clear boundary between non-STZ core and STZ core atoms as discussed. While it is noted in the previous version, to clarify this point, we have added the following sentence to the revised manuscript.

Changes in the manuscript:

(Marked manuscript: Lines 27-30)

Using our novel “frozen-atom analysis” of the simulation data, we reveal that anelastic deformation in CuZr metallic glasses is fundamentally driven by cooperative atomic motions of **tens of atoms** elastically linked to one another, forming trigger groups.

(Marked manuscript: Lines 81-83)

We found that the STZ core consists of **tens of atoms**, which engage in cooperative motion. **This value may differ among various glass systems; nevertheless, the frozen-atom analysis provides an unambiguous determination of it.**

(Marked manuscript: Lines 331-334)

This number may vary across different glass systems, and could depend on chemical composition and preparation condition. However, the frozen-atom analysis can determine this number without ambiguity.

Reference:

Zhang, Z., Ding, J., & Ma, E. (2022). Shear transformations in metallic glasses without excessive and predefinable defects. *Proceedings of the National Academy of Sciences*, 119(48), e2213941119.

Q3-3:

The technique of frozen analysis is fine, but the novelty of this method is somehow question. In the literature, Ning Xu (<https://doi.org/10.1073/pnas.2304974120>) has proposed a concept of “stabilizing bonds” that is quite related to instability of amorphous solids, which is quite similar to the frozen atom notion here. It is informative if the author could discuss them together.

A3-3:

We thank the reviewer for this valuable comment. As the reviewer points out, and as we have also emphasized in A1-9 of this reply, it is important to distinguish our frozen-atom analysis from other approaches that constrain atomic displacements, by clarifying what is fundamentally different. To address the reviewer’s suggestion, we have added a discussion in the Supplementary Materials comparing our method with the related approaches and highlighting the novelty of the frozen-atom analysis.

Changes in the manuscript:

(Marked manuscript: Lines 120-124)

This method appears similar to the artificial manipulations such as atomic pinning approach to probe atomic cooperativity in flow[24, 25] or other related interventions[26, 27], but our approach is quite different in that

atomic freezing is used merely to probe the role of each atom in deformation, and it does not affect the flow itself (see Supplementary Materials for details).

(Supplementary Materials, in Section " Frozen-atom analysis and artificial perturbations ")

Before the development of frozen-atom analysis, several studies introduced artificial manipulations to probe the physics of disordered structures. Pinning is one such approach: Berthier, *et al.* constrained the motion of randomly selected atoms under NVT conditions and thereby revealed hidden static order (point-to-set correlations) in glass forming liquids[2]. Bhowmik, *et al.* applied pinning during AQS deformation of a model amorphous solid, showing that frozen atoms suppress avalanche-like plastic events[3]. Barbot, *et al.* instead constrained the surroundings of a local region inside model amorphous solids and imposed affine deformation to forcibly trigger STZs, providing a map of local yielding propensity[1]. Xu, *et al.* proposed to remove the contribution of specific pair bonds from the force constant matrix to identify those whose elimination directly triggers instabilities[5].

Frozen-atom analysis likewise employs artificial manipulation, but differs from pinning method in that it does not suppress deformation indiscriminately throughout the AQS process. Compared with Barbot's method, the distinction lies in whether the constraint is applied inside or outside the STZ, while Xu's approach identifies critical bonds related to Local Configurational Excitation (LCE) rather than cooperative cores. The essential novelty of frozen-atom analysis is its ability to extract cooperative motion directly, a feature not accessible by previous methods. This indicates that frozen-atom analysis and these methods are complementary, and their combination may yield further insights into glass physics.

Q3-4:

What is the meaning of 'N' in equation (1). It is not clear whether it is the number of atoms in the simulation box, or the STZ core region.

Q3-5:

The abbreviation "local configurational excitations (LCEs)" has been defined a couple of times.

Q3-6:

It is not clear how large is the strain step used in the athermal quasistatic shear, which might slightly affect the definition of STZ core.

A3-4, 3-5, 3-6:

We appreciate the reviewer's attention to these points. In response, we have clarified in the revised manuscript that in equation (1) N denotes the total number of atoms in the simulation box (Q3-4). We have also removed the redundant definitions of "local configurational excitations (LCEs)" (Q3-5). Regarding the strain increment in the athermal quasistatic shear (Q3-6), as noted in A1-7 we have corrected the step size from 5×10^{-5} to 1×10^{-5} , recomputed the results with this more appropriate value, and revised the manuscript accordingly.

Changes in the manuscript:

(Marked manuscript: Line 114)

... and N denotes the total number of atoms in the simulation box.

(Marked manuscript: Lines 291-292)

This suggests that LCEs and ...

(Marked manuscript: Lines 313-314)

Each STZ core involves atomic bond rearrangement, specifically LCE, which generates...

(Marked manuscript: Lines 367-369)

This procedure was repeated 15,000 times to induce deformation until the engineering shear strain reached 15%, with a strain increment of 10^{-5} per step.

Other Changes:

Independent of the reviewers' comments, we have made broad textual and stylistic improvements throughout the manuscript. These edits are documented in the manuscript text file with track changes, and therefore we do not enumerate them in detail here.